biomaterials/materials science/nanotechnology

hypoxic drug carrier, self-amplified effect, chemoradiotherapy co-enhancement

**Authors for correspondence:**
Rui Xiong
e-mail: xiongrui@whu.edu.cn
Conghua Xie
e-mail: chxie_65@whu.edu.cn
Hong Quan
e-mail: 00007962@whu.edu.cn

# Development of a hypoxic nanocomposite containing high-Z element as 5-fluorouracil carrier activated self-amplified chemoradiotherapy co-enhancement

Cui Yang[1,†], Shan Peng[2,3,†], Yingming Sun[2,3], Hongtao Miao[4], Meng Lyu[1], Shijing Ma[2,3], Yuan Luo[2,3], Rui Xiong[1], Conghua Xie[2,3] and Hong Quan[1]

[1]Key Laboratory of Artificial Micro- and Nano-Structures of the Ministry of Education and Center for Electronic Microscopy and Department of Physics, Wuhan University, Wuhan 430072, People's Republic of China
[2]Department of Radiation and Medical Oncology, Hubei Key Laboratory of Tumor Biological Behaviors, Hubei Cancer Clinical Study Center, and [3]Center for Medical Science Research, Zhongnan Hospital of Wuhan University, Wuhan 430072, People's Republic of China
[4]Wuhan Taikang Hospital, Wuhan 430400, People's Republic of China

 CY, 0000-0002-5880-5130; HQ, 0000-0003-3510-3854

The synergetic effect of chemoradiotherapy achievement is encouraging but significantly hampered by the prevalence of hypoxia, leading to drug/radiation resistance in solid tumours. To address the problem and improve the efficiency of cancer therapy, a lamellar-structure multifunctional graphene oxide (GO) drug-delivery system with an average size of 243 nm, co-delivering of metronidazole (MI), 5-fluorouracil (5-FU) and FePt magnetic nanoparticles (MNPs), was successfully designed and synthesized in the study. The integration of hypoxic drug carrier loading radiosensitizers and chemotherapeutic drugs simultaneously, combines the properties of hypoxia-sensitivity and chemoradiotherapy co-enhancement within a single nanoplatform, which is expected to provide new ideas for cancer treatment. Through *in vitro* tests, the hypoxia-sensitivity and cytotoxicity of intracellular reactive oxygen species (ROS) of the nanocomposites (NCs) were proved. Moreover, the additive

†C. Yang and S. Peng contributed equally to the manuscript.

effect between MI, 5-FU and FePt MNPs in cytotoxicity and radiation sensitization aspects is disclosed. It performs an enhanced cell proliferation inhibition and makes up a self-amplified radiotherapy enhancement system that improves radiation efficiency and cell radiosensitivity simultaneously. In conclusion, the study recommended a novel and promising multifunctional nanoplatform which performed a self-amplified effect that activated chemoradiotherapy co-enhancement.

## 1. Introduction

Cancer, as one of the most devastating diseases, is attracting mounting attention [1–3]. However, the limitations of radiotherapy and chemotherapy, the most common approaches for cancer treatment in clinics, make the single therapeutic effect far from satisfactory [4,5]. In terms of radiotherapy, high dose output is required to completely eliminate tumours, due to the low fraction of radiation energy deposition. The high dose, however, damages normal tissues simultaneously [6]. Moreover, the general toxicity caused by nonspecific chemotherapeutics is also the crux of chemotherapy [7]. Nonetheless, oxygen deficiency, a prominent feature of solid tumours [4,8], leads to drug/radiation resistance [9–11] which exacerbates the current situation as well. Hence, the idea of developing a hypoxic drug-delivery system loading anti-cancer agent and radiosensitizers simultaneously [12] to improve the efficiency of tumour chemoradiotherapy is popular with researchers [13,14].

The possibilities of nitroimidazoles being converted to hydrophilic aminoimidazoles via bioreduction under hypoxia conditions [15,16] and trapped intracellularly [17,18] make them highly attractive for use in cancer hypoxia imaging [17,19] and hypoxia-responsive drug carriers fabrication [16]. Qian *et al*. used 2-nitroimidazole (NI) for the synthesis of light-activated hypoxia-responsive nanocarrier to enhance cancer therapy [20]. Moreover, Yu *et al*. developed a hypoxia-sensitive vesicle to deliver insulin for diabetes treatment, also using NI as a hypoxic ingredient [19]. In addition, nitroimidazoles also have been widely studied as hypoxia cells radiosensitizers [21–23]. Feketeová *et al*. [23] reported that nitroimidazoles have the ability to assist the action of radical anions to cause DNA damage in the process of radiation. However, the clinical application of nitroimidazoles is limited by the high dose needed and the subsequent neurotoxicity [16,24]. Therefore, the crucial solution for employing nitroimidazoles as radiosensitizers is to transport them into the tumour area to minimize the cytotoxicity of normal tissue [16].

Recent developments in nanotechnology provide great opportunities for biomaterials fabrication [25,26]. Especially, graphene oxide (GO), the oxidation derivative of graphene, a two-dimensional lamellar structure nano-system of one atom thickness [27,28], attracts interest owing to its specific surface area and excellent biocompatibility [29]. Moreover, it has been reported that PEGylated GO can effectively avoid the risk of being aggregated in solutions rich in salts and proteins [30,31].

In the study, a PEGylated multifunctional magnetic graphene oxide (MGO) drug carrier system, co-delivery of metronidazole (MI) and 5-fluorouracil (5-FU) [32,33], was successfully designed and synthesized, designated as MGO/FU-MI nanocomposites (NCs). The design objectives are as follows: (i) PEGylated GO with MI covalently bonded, makes up the hypoxic drug carrier delivering 5-FU into the tumour site. (ii) The high-Z material [34], FePt magnetic nanoparticles (MNPs) [35,36], combined with MI constituting the radiosensitization system to enhance radiotherapy efficiency. (iii) Co-delivery of radiosensitizers and chemotherapeutics to achieve the synergetic effect of chemoradiotherapy. Furthermore, it was hypothesized that the existence of MI would guide more NCs trapped in tumour cells due to their hypoxia response characteristics, which leads to enhanced growth inhibition of cells under hypoxia conditions. In addition, 5-FU hinders sublethal damage repair, which significantly improves radiotherapy efficiency at the biological level. Thus, the co-delivery of MI and 5-FU can produce an additive effect both in cytotoxicity and radiotherapy aspects that activate chemoradiotherapy co-enhancement.

## 2. Experimental section

### 2.1. Materials

Chloroplatinic acid ($H_2PtCl_6 \cdot 6H_2O$, Reagent No.1 Factory Of Shanghai Chemical Reagent Co., Ltd., Shanghai, China; analytical reagent), sodium borohydride ($NaBH_4$, Sinopharm Chemical Reagent Co.,

Ltd., Shanghai, China; 96%), polyethylene glycol 2000 (PEG, Sinopharm Chemical Reagent Co., Ltd., Shanghai, China; guarantee reagent), anhydrous ethanol (Sinopharm Chemical Reagent Co., Ltd., Shanghai, China; analytical reagent), iron acetylacetonate (Fe(acac)$_3$, Aladdin Industrial Corporation, Shanghai, China; 98%), oleic acid (C$_{18}$H$_{34}$O$_2$, OA, Aladdin Industrial Corporation, Shanghai, China; analytical reagent), oleylamine (C$_{18}$H$_{37}$N, OL, Aladdin Industrial Corporation, Shanghai, China; 95%), 1-ethyl-(3,3-dimethylaminopropyl carbodiimide) (EDC.HCl, Shanghai Medpep Co., Ltd., Shanghai, China, 99%), 4-dimethylaminopyridine (DMAP, Shanghai Titan Scientific Co., Ltd., shanghai, China, 99%), graphene oxide (GO, Suzhou tanfen tech. Co., Ltd., Suzhou, China) solution 2-(2-methyl-5-nitro-1H-imidazol-1-yl), ethanol (MI, Shanghai Bioengineering Co., Ltd., Shanghai, China, Reagent grade), 5-fluorouracil (5-FU, Tianjin xinyao pharmaceutical Co., Ltd., Tianjin, China), 3-(4,5-dimethylthiazol-2-yl)-2, 5-diphenyltetrazolium bromide (MTT, Shanghai yuanye biotechnology Co., Ltd., Shanghai, China), Cell cycle detection Kit (Jiangsu KeyGEN BioTECH Co., Ltd., Jiangsu, China), Reactive Oxygen Species Assay Kit (Beyotime Biotech Co., Ltd., Shanghai, China).

## 2.2. Nanocomposites synthesis

### 2.2.1. Preparation of FePt MNPs

FePt MNPs were prepared by chemical co-reduction method as previously described [7,13,36]. Briefly, OA (1.5 ml), OL (1.5 ml) and Fe(acac)$_3$ (0.386 mmol) were dissolved in anhydrous ethanol (100 ml) and stirred for 30 min. Then, H$_2$PtCl$_6$·6H$_2$O ethanol solution (20 ml, 19.3 mmol l$^{-1}$, 0.386 mmol) was transferred into the mixture mentioned above and stirred for another 30 min. Whereafter, NaBH$_4$ ethanol solution (65.7 mmol l$^{-1}$, 200 ml) was added drop by drop at a temperature of 40°C and stirred for an hour. Subsequently, the black product was separated by centrifugation (7500 rpm, 5 min) and dried in vacuum at 40°C overnight.

### 2.2.2. Preparation of GO-MI NCs

The PEG (0.25 mmol) was dissolved in 100 ml of deionized water. Next, GO (5 ml, 5 mg ml$^{-1}$), EDC (3.5 mmol) and MI (3.15 mmol) was added and mixed thoroughly. Then DMAP (7.0 mmol) was added drop by drop at room temperature and stirred vigorously for 72 h. The brownish product obtained by centrifugation (8000 rpm, 5 min) was designated as GO-MI NCs.

### 2.2.3. Preparation of MGO-MI NCs

FePt MNPs (50 mg) and GO-MI NCs (80 mg) were dissolved in 150 ml deionized water in a 250 ml florence flask. After sonication at room temperature for 6 h, the black product was obtained by centrifugation (12 000 rpm, 10 min) and dried in vacuum at 40°C overnight.

### 2.2.4. Preparation of MGO/FU-MI NCs

FePt MNPs (50 mg) and GO-MI NCs (80 mg) were dissolved in 150 ml deionized water in a 250 ml florence flask. 5-FU (20 ml, 2 mg ml$^{-1}$) was added to the mixture. The mixture was sonicated for 6 h at room temperature, the black product was obtained by centrifugation (12 000 rpm, 10 min) and dried in vacuum at 40°C overnight.

### 2.2.5. Preparation of MGO/FU NCs

The MGO/FU NCs were prepared by the same process as MGO/FU-MI NCs without adding the reagent of MI in the process of GO-MI NCs synthesis.

## 2.3. MGO/FU-MI NCs characterization

The observations of the size, morphology and dispersibility of samples were on a JEM2010FEF-Ω (ultra-high resolution, UHR) transmission electron microscope (TEM, JEOL, Japan) and a dynamic light scattering device (DLS, Zetasizer Nano ZSP). Surface chemical compositions analyses of samples were examined on energy dispersive X-ray spectroscopy (EDS) and X-ray photoelectron spectroscopy (XPS, ESCALAB250Xi, Thermo Fisher). The chemical structure was examined by Fourier transform infrared spectroscopy (FT-IR) spectrometer (Nicolet 6700, Thermo Fisher, USA) with a resolution of 5 cm$^{-1}$.

## 2.4. Drugs loading efficiency and releasing curves

The loading efficiency and releasing curves of MI and 5-FU in the MGO/FU-MI NCs were determined using UV–vis spectrophotometer (Lambda 650S, PerkinElmer). Firstly, a series of deionized water solutions with different MI and 5-FU concentrations were prepared to obtain the calibration curve. Then, the absorbances of 5-FU and MI of the solutions at the beginning and end of the synthesis were detected to calculate the reduced parts, which corresponded to the amount loaded on the NCs. And the loading efficiency and capacity was calculated using the following formulae [16]:

$$\text{Loading efficiency (\%)} = \left( \frac{\text{the weight of loading drug}}{\text{the weight of drug to feed}} \right)$$

and

$$\text{Loading capacity (\%)} = \left( \frac{\text{the weight of loading drug}}{\text{the weight of NCs}} \right).$$

The loading efficiencies and capacity were 25.03% and 12.3% for MI and 27.76% and 9.5% for 5-FU on MGO/FU-MI NCs respectively in the present synthesis. Moreover, the releasing profiles of MI and 5-FU were obtained through detecting the absorbances of PBS solutions with different PH, which contained 500 μg ml$^{-1}$ NCs and were incubated for 1, 3, 6, 12, 24 h.

## 2.5. Cell culture and treatment

Adenocarcinoma human alveolar basal epithelial cells (A549, Type Culture Collection of the Chinese Academy of Sciences, Shanghai, China) and Human lung adenocarcinoma cells (H1975, Type Culture Collection of the Chinese Academy of Sciences, Shanghai, China) were used in this study. Both of them have passed the STR authentication by Guangzhou Cell cook Biotech Co., Ltd. Both cell lines were cultured in RPMI-1640 medium (HyClone, USA) containing 10% fetal bovine serum (FBS, HyClone), 100 units antibiotics penicillin per ml and 100 μg streptomycin per ml (Beyotime Biotechnology, Shanghai, China). Cells culture at 37°C incubator (Sanyo Electric Co., Ltd., Japan) with 5% $CO_2$ which provides normoxia conditions. The hypoxia situation (p$O_2$: 5%) was achieved in a hypoxia incubator (MCO-170MUVL, Panasonic, Japan) with a mixture of 5% $O_2$, 5% $CO_2$ and 90% $N_2$. Cells were washed by phosphate buffer saline (PBS) and digested by 0.25% trypsin and 0.02% EDTA solution (Genom biomedical technology Co., Ltd., China).

## 2.6. Cell viability assay

The cell proliferation inhibition of A549 and H1975 cells under normoxic and hypoxic situation was determined by MTT assay. Take H1975 cells as an example: logarithmic phase cells were seeded in 96-well plates with 8000 cells in each well and cultured overnight for cell attachment. Then different concentrations of MGO-MI (MGO/FU or MGO/FU-MI) NCs solutions were added into the wells. After 24 h incubation, 20 μl MTT (5 mg ml$^{-1}$) was added to each well and incubated for an additional 4 h. The mixture was then removed slightly and 150 μl DMSO was added to each well for 15 min to dissolve purple formazan. Finally, quantitative measurements (absorbance) were obtained at the wavelength of 570 nm using a multifunctional enzyme marking instrument (PE Enspire). The cell viability is calculated using the follow equation [7]:

$$\text{cell viability} = \frac{A - A_c}{A_0 - A_{c0}},$$

where $A$ was the absorbance of experimental group, $A_c$ represents the absorbance of non-seeded group, $A_0$ represents the absorbance of control group, $A_{c0}$ was the absorbance of RPMI-1640 medium. The results were analysed by GraphPad Prism software (Beckman Coulter, Inc.) and expressed as means ± standard deviation (s.d.).

## 2.7. Flow cytometric analysis

### 2.7.1. Cell-cycle analysis

To evaluate the cell-cycle regulation effects of NCs, cells were seeded into 6-well plates (3 × 10$^5$ per well) and cultured overnight. Then cells were treated with MGO-MI, MGO/FU, MGO/FU-MI NCs solutions

with final concentrations of 20 µg ml$^{-1}$ for each well respectively, and incubated for 24 h. Next, cells were washed, trypsinized, centrifuged and resuspended in 300 µl PBS. After that, cells were fixed with 700 µl pre-cooled anhydrous ethanol added drop by drop, and placed at −20°C refrigerator overnight. Fixed cells were washed, centrifuged and resuspended in 500 µl PI/RNase A solution (RNase A : PI = 1 : 9), and then incubated for 30 min at 37°C without light. At the end of the treatment, the cell-cycle regulation was detected by flow cytometry (Cytomics TM FC 500, Beckman Coulter, Inc., USA) and 10 000 events were counted for each sample. CytExpert, FlowJo and GraphPad Prism software were used to analyse the data. The results were expressed as mean ± s.d.

### 2.7.2. Intracellular reactive oxygen species generation

The effects of NCs on intracellular reactive oxygen species (ROS) generation were examined by ROS assay kit (Beyotime, China), which uses a dichloro-dihydro-fluorescein diacetate (DCFH-DA) fluorescence probe to detect intracellular ROS [13,37]. Specifically, DCFH-DA itself has no fluorescence and can pass through the cell membrane freely. After entering the cell, it can be hydrolysed to DCFH by intracellular esterase, which cannot penetrate the cell membrane, making it easy for the probe to be loaded into the cell. Moreover, ROS in the cells can oxidize non-fluorescent DCFH and produce fluorescence DCF [38,39]. Therefore, the level of intracellular ROS could be known through measuring the fluorescence intensity of DCF in cells [40]. In the experiments, $3 \times 10^5$ cells were seeded in 6-well plates and cultured overnight. Then cells were treated with MGO-MI, MGO/FU, MGO/FU-MI NCs solutions with final concentrations of 20 µg ml$^{-1}$ for each well, and incubated for 24 h. Next, cells were washed, trypsinized, centrifuged and resuspended in 10 mM DCFH-DA serum-free medium solution and incubated for 30 min at 37°C without light. After that the cells were washed twice with serum-free medium and collected for fluorescence analysis using a flow cytometer (Cytomics TM FC 500, Beckman Coulter, Inc., USA) and 10 000 events were counted for each sample. The data were analysed using FlowJo and GraphPad Prism software. The results were expressed as means ± s.d.

## 2.8. Clonogenic survival assays

### 2.8.1. Clonogenic survival assay without radiation

A series of clonogenic survival experiments were conducted on MGO/FU-MI NCs with different concentrations (0, 10, 15, 20 µg ml$^{-1}$) in H1975 and A549 cells under non-radiation conditions to determine the optimal concentration for radiation sensitivity examination [12]. Take H1975 cells as an example: H1975 cells were seeded in two 6-well plates (200 cells per well) and incubated overnight. Then four concentrations of 0, 10, 15, 20 µg ml$^{-1}$ MGO/Fu-MI NCs were added, each containing three wells, and incubated for about 15 days until the cell clones could be observed with the naked eyes. Next, cells were washed twice with PBS, fixed with 4% paraformaldehyde (Biosharp, China) for 30 min, and stained with 0.5% crystal violet for 20 min. Finally, the number of cell clones in each well was counted. The planting efficiency (PE) and survival fraction (SF) of the clones were used to evaluate the effect of different treatments, the equations for which are as follows [4,16]:

$$PE = \frac{\text{the number of clones in control group}}{\text{the number of cells planted in control group}}$$

and

$$SF = \frac{\text{the number of clones in experimental group}}{\text{the number of cells planted in experimental group}^* \, PE}.$$

### 2.8.2. Clonogenic survival assay under 2Gray (Gy) radiation

After determining the experimental concentration, clonogenic survival assays under 2 Gy radiation, the most common clinical dose fractionation, were conducted to examine the radiation enhancement effect of MGO/FU-MI NCs preliminarily. Briefly, H1975 (or A549) cells was seeded in two 6-well plates with 200 and 500 cells pre well for each plate and incubated overnight. Then the optimal experimental concentration of NCs was added to three wells of the plate with 500 cells per well and incubated for another 4 h. Then a Siemens Primus HI Linear Accelerator (Zhongnan Hospital of Wuhan University, Wuhan, Hubei, China) was used to conduct 2 Gy X-ray irradiation on the plates of 500 cells per well

with a 6 MV photon beam. The radiation field was $35 \times 35$ cm at a source−surface distance of 100 cm. The ongoing process and data analysis were the same as above. Moreover, the SF under 2 Gy ($SF_2$) was calculated using the equations above [4].

### 2.8.3. Clonogenic survival assay under various doses of radiation

Clonogenic survival assays under various doses of radiation were conducted to demonstrate the radiation sensitization effect of MGO/FU-MI NCs. Analogously, H1975 (or A549) cells were seeded in seven 6-well plates with the number of cells per well in each plate being 200, 300, 500, 800, 2000, 5000 and 10 000, respectively. After cells adherence, the optimal experimental concentration of MGO/Fu-MI NCs were added into three wells of each plate and incubated for 4 h. Then, the plates were irradiated with X-ray for 0, 1, 2, 4, 6, 8 and 10 Gy respectively. The following process and data analysis were the same as above.

In addition, the survival curve was fitted with the 'multi-target single hitting' model, the equation for which is as follows [4,13]:

$$\text{SF} = 1 - \left( 1 - \exp\left( \frac{-D}{D_0} \right) \right)^N,$$

where $D$ is the irradiation dose while $D_0$ and $N$ are the parameters representing the mean lethal dose and the target spot number in DNA respectively in radiobiology [41,42], which were obtained through function fitting by GraphPad Prism software. Moreover, the radiotherapy enhancement effect was evaluated using dose enhancement factor (DEF) at 90% survival fraction [43,44] and sensitization enhancement ratio (SER) [4,16], the equation for which is as follows:

$$\text{DEF} = \frac{D_c(SF_{90})}{D_d(SF_{90})}; \quad \text{SER} = \frac{D_{0c}}{D_{0d}},$$

where $D_c(SF_{90})$ represented the dose producing 90% SF for the control group, while the $D_d(SF_{90})$ was the dose producing 90% SF for the drug group. Similarly, $D_{0c}$ and $D_{0d}$ were the parameters $D_0$ obtained in the control group and the drug group respectively. The results were normalized and expressed as means $\pm$ s.d.

## 2.9. Statistical analyses

Each experiment was carried out three times and the data were presented in the form of three individual experiments. A two-tailed Student's $t$-test was used to evaluate the statistical significance of each group. Statistical analysis was conducted using GraphPad Prism software. $*p < 0.05$ and $**p < 0.01$ were considered statistically significant.

# 3. Results

## 3.1. Synthesis and physical characterization of MGO/FU-MI NCs

The synthesis of MGO/FU-MI NCs was accomplished step by step (figure 1*a*). Firstly, GO was modified by MI and PEG through an EDC chemistry [16,31,45] and designated as GO-MI NCs. The FePt MNPs were prepared by a chemical co-reduction method described in our previous work [7,13,36]. Secondly, FePt MNPs and 5-FU were loaded onto GO-MI NCs via a water-bath ultrasonication process. Finally, the MGO/FU-MI NCs were successfully obtained through centrifugation and desiccation. The characters of the NCs were detected by a series of physical characteristic measurements, including morphology, size and chemical composition (figure 2).

The morphology, dispersion and size of MGO/FU-MI NCs were characterized by transmission electron microscope (TEM). As figure 2*a* exhibits, the lamellar structure of GO with a size of nearly 250 nm can be observed clearly. Moreover, the FePt MNPs with spherical morphology are uniformly layered on the surface of GO with high loading ratio. In figure 2*b*, the FePt MNPs can be seen grain by grain at a higher resolution, suggesting the monodispersity of FePt MNPs. Besides, the average diameter of MGO/FU-MI NCs is about 243.6 nm, analysed with dynamic light scattering (DLS) (figure 2*c*). In addition, X-ray photoelectron spectroscopy (XPS) and energy dispersive X-ray spectroscopy (EDS) were conducted to determine the chemical composition of the NCs. Figure 2*d* shows the XPS spectra of GO and GO-MI NCs. The peak of

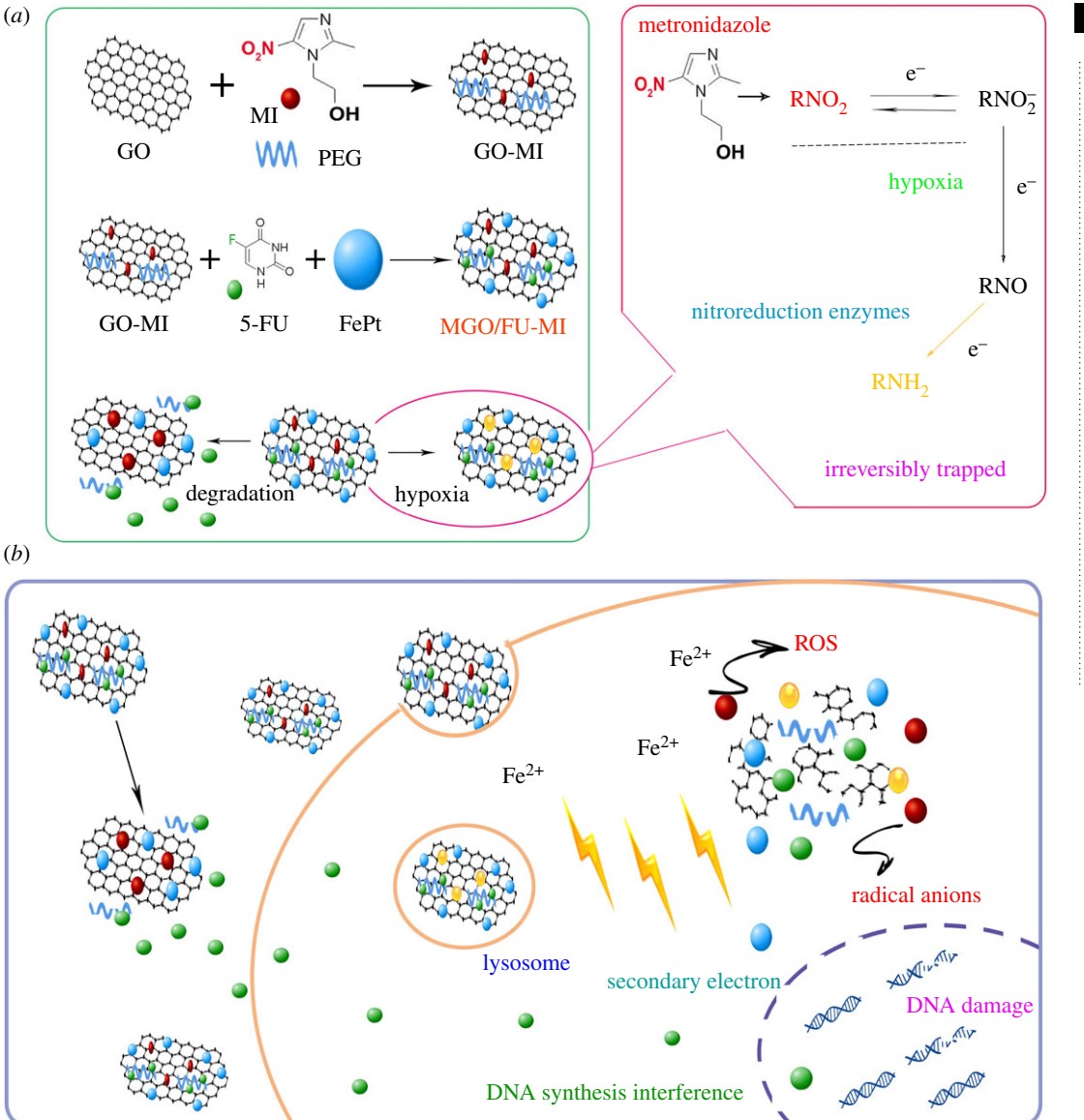

**Figure 1.** Schematic diagram of hypoxic drug delivery system. (*a*) Formation process of MGO/FU-MI nanocomposites (NCs) and hypoxia sensitivity mechanism of metronidazole (MI). (*b*) Schematic illustration of antitumour effect of MGO/FU-MI NCs. After cellular uptake, the NCs degrades in the assistance of lysosome and releases cargos. 5-FU has the effect of DNA synthesis interference while FePt magnetic nanoparticles (MNPs) can stimulate intracellular reactive oxygen species (ROS) overproduction, which significantly suppresses cell proliferation. In addition, the FePt MNPs and MI can improve radiation energy deposition and radical anions production, while 5-FU hinders sublethal damage repair which constitutes a self-amplified radiation sensitization system that improves radiotherapy efficiency.

binding energy at 399.46 eV on the spectrum of GO-MI NCs belongs to N1s, and the atomic ratio is 6.1%, proving the existence of MI on GO after the process of GO modification. Furthermore, in the spectrum of MGO/FU-MI NCs (figure 2*e*), F1s peaks at the binding energy of 678.96 eV appearing after 5-FU is loaded, the atomic ratio of which is 3.1%, which evidences the load of 5-FU though to a slight degree. Alternatively, the EDS analysis demonstrates the conclusion once again in figure 2*f*, in which the peak of Cu comes from the copper mesh that loads samples.

The chemical structure changes of the NCs along the synthesis were determined by Fourier transform infrared spectroscopy (FT-IR) (figure 3), and the loading methods of MI and 5-FU were disclosed. In the spectrum of GO (figure 3*a*), the strong and broad peak at 3423 cm$^{-1}$ is attributed to hydroxyls and undried water molecules, while the characteristic signals at 1720 and 869 cm$^{-1}$ belong to the carbonyls in carboxyls groups and epoxy groups respectively [30,31,45]. Besides, the characteristic peaks at 1564 and 1380 cm$^{-1}$ in the spectrum of GO-MI NCs were assigned to nitro groups of MI. By

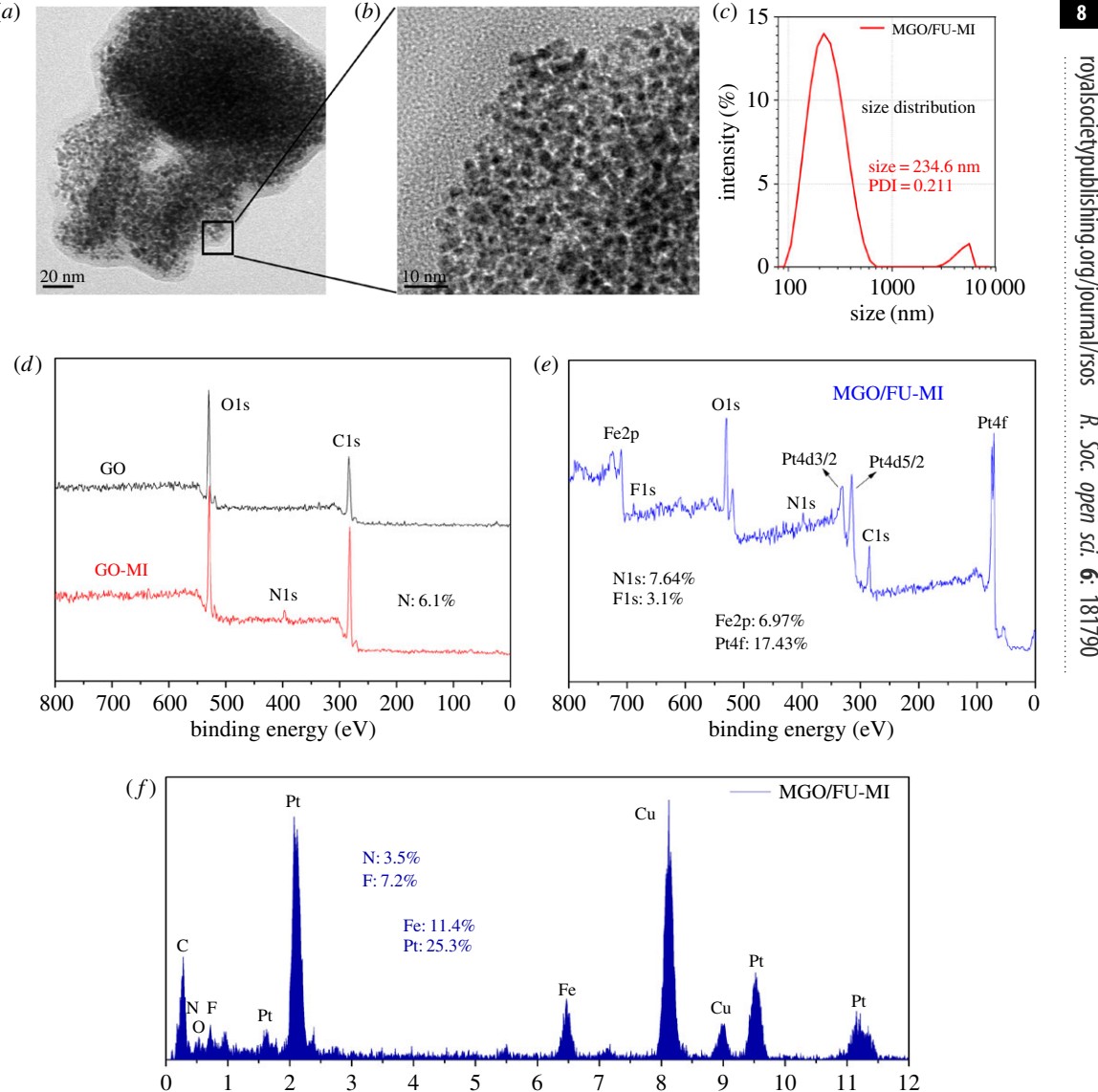

**Figure 2.** Characterization of the NCs. TEM micrographs of MGO/FU-MI NCs in scale of (*a*) 20 nm and (*b*) 10 nm. (*c*) DLS of MGO/FU-MI NCs, with an average diameter was 243.6 nm. (*d,e*) XPS spectra of GO, GO-MI, MGO/FU-MI NCs, which characterize the chemical composition changes along the synthesis. (*f*) EDS analysis of MGO/FU-MI NCs.

the way, we note that the stretching vibration peak of carboxyls at 1720 cm$^{-1}$ disappears, which demonstrates that MI and PEG covalently connect to GO via esterification of hydroxyl and carboxyl groups belonging to MI and GO respectively [15,16] (figure 3*c*). Otherwise, in the spectrum of MGO/FU-MI NCs (figure 3*b*), the enhanced characteristic peaks of methylene, which are located at 2923 and 2852 cm$^{-1}$, and the appearance of the peak of carbon−fluorine bond (C-F) at 1261 cm$^{-1}$ both verify the successful load of 5-FU on GO-MI NCs. No significant peaks disappear, suggesting that 5-FU is loaded on GO via physisorption with the support of PEG [30,45] (figure 1*a*).

In addition, the loading characteristics of the NCs were examined using a UV−vis spectrophotometer. Under the raw materials ratio of the present synthesis, the loading efficiency and capacity of MI on NCs are 25.03% and 12.3%, while those of 5-FU are 27.64% and 9.5%, respectively. The low load is consistent with the results of XPS and EDS examination. Moreover, the releasing curves of MI are illustrated in figure 3*d*. As shown, after 24 h incubation, the MI is hardly released in neutral PBS. And even in acidic PBS, its release is slight, which is 0.3 μg ml$^{-1}$. The result suggests the stability of the NCs even in the acidic environment. In comparison, after 24 h incubation, the release of 5-FU is relatively high, being 2.8 μg ml$^{-1}$ in both neutral and sub-acid environments (figure 3*e*). The difference in MI and 5-FU releasing characteristics may be connected to the different loading methods of the two drugs. The

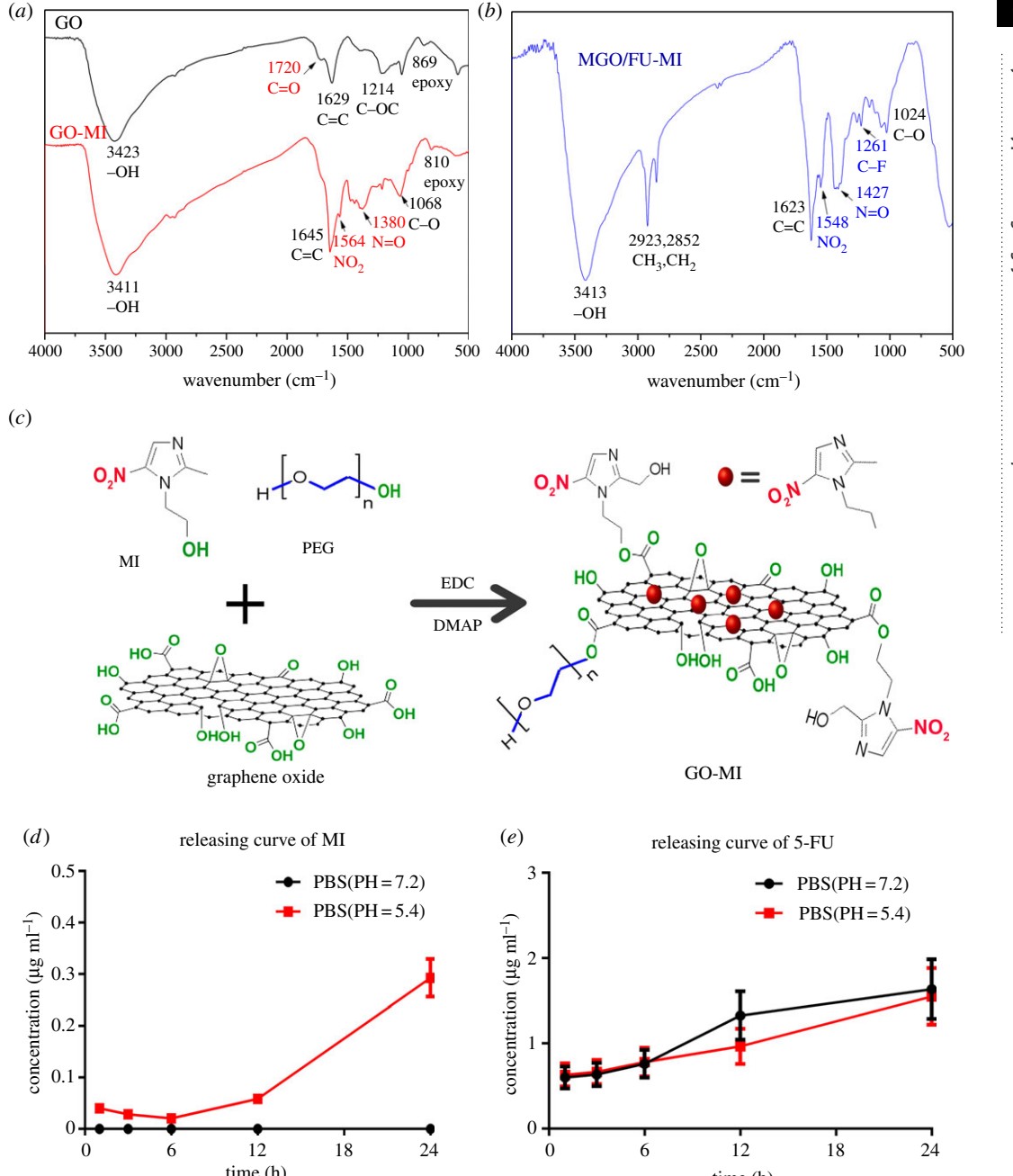

**Figure 3.** Chemical structure and drug releasing character of the NCs. The FT-IR spectra of (*a*) GO and GO-MI (*b*) MGO/FU-MI NCs, which characterize the chemical structure changes of the NCs along the synthesis. (*c*) Schematic illustration of the EDC chemistry. The releasing curves of (*d*) MI and (*e*) FU in phosphate buffer solution (PBS) with PH values of 7.2 and 5.4.

former is covalently clinked to GO, while the latter is loaded onto GO via physisorption with the support of PEG, so it is relatively easier than MI release.

## 3.2. The cell viability analyses of the nanocomposites

After 24 h incubation, the intracellular distribution of MGO/FU-MI NCs in H1975 cells was observed by TEM. As figure 4*a* shows, the NCs has been completely encapsulated in the cytoplasm, close to the nuclear envelope. Moreover, at high magnification, some NCs can be seen to have degraded and released the cargos into cytoplasm. In addition, the intact and undamaged cell membrane directly indicates the good biocompatibility of the NCs.

Moreover, MTT assay was conducted to evaluate the inhibitory effect of NCs on the proliferation of H1975 and A549 cell lines, in which MGO/FU and MGO-MI NCs were synthesized to compare the

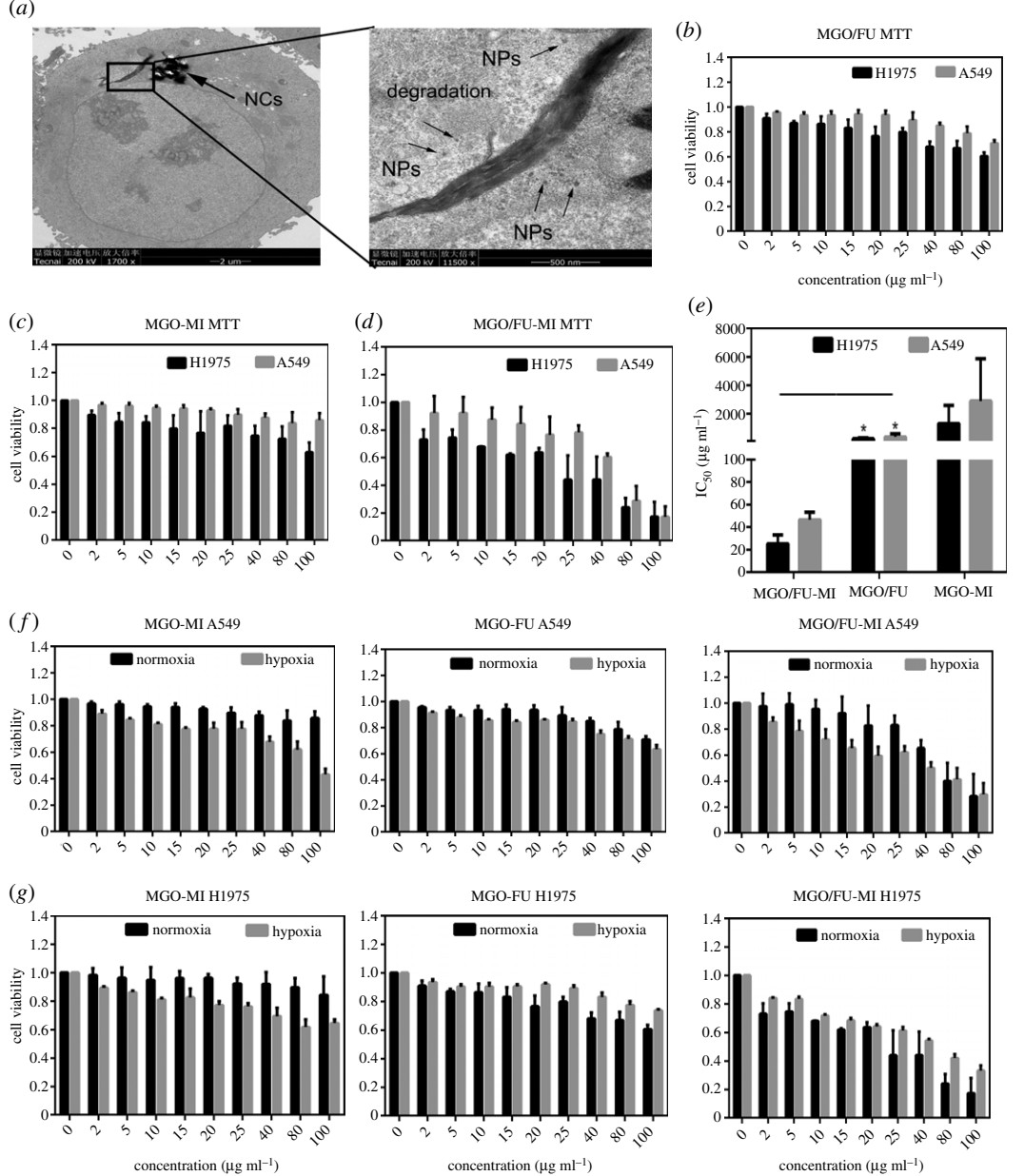

**Figure 4.** Cytotoxicity and hypoxia sensitivity of the NCs. (*a*) TEM micrographs of cell uptake and NCs degradation. Cell proliferation inhibition effect of (*b*) MGO/FU, (*c*) MGO-MI and (*d*) MGO/FU-MI NCs, evaluated by MTT assay, using H1975 and A549 cells under normoxia situation ($pCO_2$: 5%). (*e*) The $IC_{50}$ values of the NCs, expressing as mean $\pm$ s.d. of three independence experiments, $*p < 0.05$, $**p < 0.01$. The hypoxia sensitivity of the NCs determined by MTT assay using (*f*) A549 and (*g*) H1975 cells under hypoxia situation ($pO_2$: 5%, $PCO_2$: 5%, $PN_2$: 90%).

antitumour effects of the two loaded chemotherapeutics. As figure 4*b*,*c* displays, the cytotoxicity of NCs shows concentration and cell-type dependence. Among them, the inhibitory effects of MGO/FU and MGO-MI NCs at the concentrations of $2-80\ \mu g\ ml^{-1}$ are negligible, indirectly implying the commendable biocompatibility of the drug-delivery system. On the contrary, cell proliferation is significantly inhibited after MGO/FU-MI NCs treatment. As figure 4*d* shows, the inhibition rate of H1975 is nearly 30% at $2\ \mu g\ ml^{-1}$ and 50% at $25\ \mu g\ ml^{-1}$, which is nearly 50-fold higher than that of MGO/FU and MGO-MI NCs. The enhanced inhibitory effect of MGO/FU-MI NCs reveals a strongly additive effect between MI and 5-FU directly. Notably, when it comes to A549 cell lines, the inhibitory effect of these three NCs is relatively weak, indicating the cytotoxicity of the NCs in a cell-dependent manner. Additionally, the inhibition concentration of 50% ($IC_{50}$) of the three NCs was calculated and depicted in figure 4*e*. It can be seen that the $IC_{50}$ for H1975 and A549 cells treated by MGO/FU-MI NCs were 25 and $46\ \mu g\ ml^{-1}$ respectively, which are much smaller than those of the

other two NCs. It directly displays the astonishing enhancement effect of MGO/FU-MI NCs on cell proliferation inhibition and emphasizes the additive effect again.

Furthermore, the hypoxia sensitivity of NCs was evaluated under hypoxia condition at 5% oxygen concentration. As shown in figure 4f,g, compared with normoxia condition, both H1975 and A549 cells cultured by MGO-MI NCs show decreased cell viability under hypoxia condition. As calculated, the drug cytotoxicity of MGO-MI NCs is increased by nearly 50 times under hypoxia condition, demonstrating the hypoxia sensitivity of MI. As illustrated in figure 1a, MI can be converted and trapped intracellularly under hypoxia condition, leading to drug enrichment in hypoxic cells [15,17], thus enhancing cytotoxicity. Nevertheless, when it comes to MGO/FU NCs, the results are cell-type dependence. It can be seen that the cell viability of A549 cells is decreased after MGO/FU NCs treatment while that of H1975 cells isn't. This phenomenon occurs because hypoxia can stir up diverse cellular regulatory mechanisms, influencing cell uptake and metabolism, resulting in different levels of drug resistance in different cell lines [46]. Regarding MGO/FU-MI NCs, the results are interesting. Firstly, compared with the other two NCs, the inhibitory effect of MGO/FU-MI NCs is still enhanced under whatever conditions, implying the strong additive effect between MI and 5-FU. Additionally, the decrease in cell viability under hypoxia is observed only in A549 cells. It implies that the cytotoxicity of MGO/FU-MI NCs is a result of combined actions of cellular regulatory mechanism under hypoxia condition together with additive effect between MI and 5-FU on pharmacokinetics.

## 3.3. The cell cycle regulation and ROS augment effect of nanocomposites

Exposure to the three NCs at concentration of 20 µg ml$^{-1}$ for 24 h, cell cycle regulation and intracellular ROS levels were examined by flow cytometry analysis to further explore the antitumour mechanism of MGO/FU-MI NCs and define the biological effect of each drug (figure 5). As shown in figure 5a, the three NCs perform different biological effects on cell cycle regulation. Compared with the control group, MGO/FU NCs shows an S phase arresting effect, the fraction of which is almost one-time enhanced on H1975 cells. This is due to the DNA synthesis interference effect of 5-FU. In the past 50 years, 5-FU have been extensively studied and applied as anti-metabolites [33,47]. Longley et al. have reported in the journal Nature in 2003 [48] that the misincorporation of fluoronucleotides into RNA and DNA and the inhibition of the action of thymidylate synthase (TS) contribute to the DNA synthesis or repair interference effect of 5-FU [49,50]. Thus, the S phase arresting effect of MGO/FU NCs is understandable. Besides, MGO-MI NCs displays a slight augmentation of the G2 phase, suggesting the cell radiosensitivity improvement effect of MI [23]. Moreover, the cell cycle regulation effect of MGO/FU-MI NCs exhibits the characteristics of the two drugs, which again indicates the additive effect of the two drugs from the biological perspective. Figure 5b,c depicts the statistical results of three independent cell cycle analyses of H1975 and A549 cells respectively, which are consistent with the conclusions mentioned above, and demonstrate that the differences are statistically significant.

Intracellular ROS, such as hydroxyl radicals, hydrogen peroxide and superoxide anions [51,52], show destructive effects on DNA and proteins. Its overproduction can affect cell viability and even cause cell death through necrosis or apoptosis [53–55]. The results of intracellular ROS detection of H1975 cells under different treatment were depicted in figure 5d. Compared with the control group, the fluorescence intensity histograms of MGO/FU and MGO-MI NCs treatment groups are slightly shifted to the right, indicating that the number of cells with high fluorescence intensity increases after NCs treatment. In other words, the intracellular ROS increase slightly after MGO/FU and MGO-MI NCs treatment. It may due to the effect of FePt MNPS. Many researchers have found that FePt MNPs can lead to ROS increment through Fenton reaction [13,56]. Therefore, the ROS augmentation after NCs treatment is understandable. Moreover, the right shift of fluorescence intensity histogram of the MGO/FU-MI group is obvious, which suggests that the ROS generation effect of the NCs comes not only from the FePt MNPs, but also from the additive effect between MI and 5-FU. As figure 5e shows, the ROS augment results of A549 cells are similar but to a slight degree, which is consistent with the MTT assays. In addition, the statistical results of three separate ROS detections on H1975 and A549 cells are shown in figure 5f,g, statistically significant enhancement exhibits both in the mean and median fluorescence intensity after NCs treatment.

## 3.4. The radiotherapy enhancement effect of the nanocomposites

The clonogenic survival assays under 0 and 2 Gy radiation were carried out respectively to determine the appropriate experimental concentration and the radiation enhanced ratio under the clinical conventional

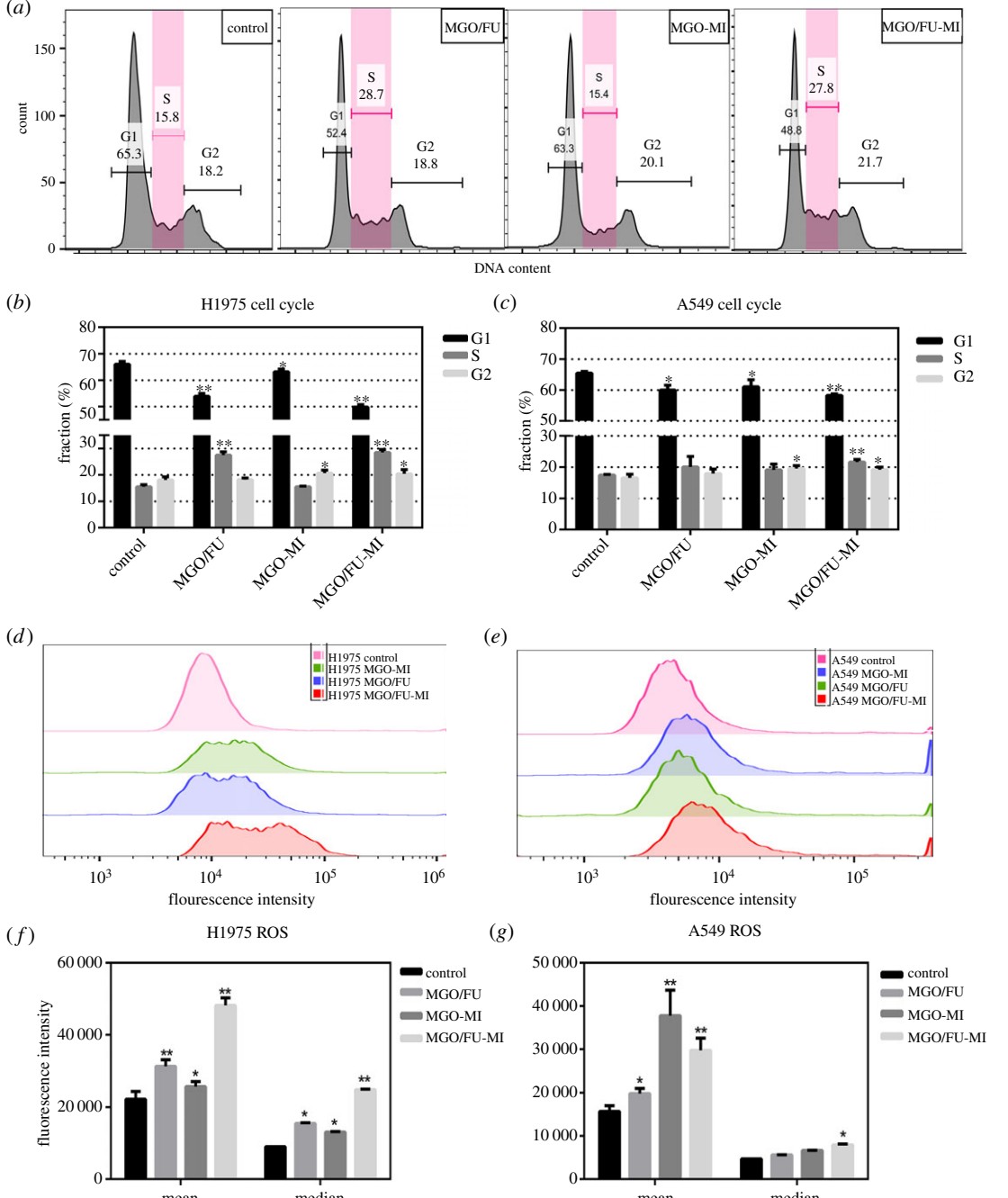

**Figure 5.** The antitumour mechanism analyses of the NCs. (a) One of the results of cell cycle regulation analysis of H1975 cells treated by MGO/FU, MGO-MI and MGO/FU-MI NCs at 20 $\mu$g ml$^{-1}$. Statistical analysis of cell cycle regulation of (b) H1975 and (c) A549 cells of three independent experiments, the values ware expressed as mean $\pm$ s.d. *$p < 0.05$, **$p < 0.01$. The intracellular ROS level of (d) H1975 and (e) A549 cells treated by MGO/FU, MGO-MI and MGO/FU-MI NCs at 20 $\mu$g ml$^{-1}$. Statistical analysis of intracellular ROS level of (f) H1975 and (g) A549 cells of three independent experiments, the values ware expressed as mean $\pm$ s.d., *$p < 0.05$, **$p < 0.01$.

fractional dose. As figure 6a shows, after exposure to the NCs for 15 days, the rate of formation of clones decreases as concentration increasing for both cell types. The concentrations of 20 $\mu$g ml$^{-1}$, used for cell cycle and ROS generation detection, is too toxic to be used as experimental concentration for clongenic survival assay. Therefore, the concentration of 15 and 10 $\mu$g ml$^{-1}$ were selected for H1975 and A549 cells respectively to be used in the later clongenic survival assay with radiation. Moreover, as is depicted in figure 6b, the SF under 2 Gy (SF$_2$) decreases significantly after the NCs treatment. It changes from 0.58 to 0.36 for H1975 cells and 0.79 to 0.52 for A549 cells. Through calculating, the radiation enhanced ratios are 1.6 and 1.5 respectively, which suggests a great application prospects in clinics.

**Figure 6.** The radiotherapy enhancement effect of NCs. (*a*) The survival fraction (SF) of H1975 and A549 cells treated with different concentration of MGO/FU-MI NCs without radiation. (*b*) The SF$_2$ (the SF under 2 Gy) of the H1975 cells treated 15 μg ml$^{-1}$ NCs and A549 cells treated with 10 μg ml$^{-1}$ NCs. The survival curve fitted by multi-target single hitting model of (*c*) H1975 treated with 15 μg ml$^{-1}$ NCs and (*d*) A549 cells treated with 10 μg ml$^{-1}$ NCs and the EDFs at 90% survival ratio.

**Table 1.** Table of $D_0$, $N$, $D_q$, SF$_2$ and SER values for cells with radiotherapy. $D_0$: mean lethal dose; $N$: the number of target; $D_q$: prospective domain dose; SF$_2$: the survival fraction at 2 Gy; SER: sensitivity enhancement ratio.

| cell lines | treatment | $D_0$ (Gy) | $N$ | $D_q$ (Gy) | SF$_2$ | SER |
|---|---|---|---|---|---|---|
| H1975 | control | $2.873 \pm 0.4981$ | $1.277 \pm 0.2764$ | 3.6688 | 0.586 | 1.152 |
| | MGO/FU-MI | $2.493 \pm 0.7577$ | $0.8175 \pm 0.2716$ | 2.0380 | 0.385 | |
| A549 | control | $2.878 \pm 0.4824$ | $1.871 \pm 0.4649$ | 5.3847 | 0.726 | 1.998 |
| | MGO/FU-MI | $1.440 \pm 0.3179$ | $3.358 \pm 1.322$ | 4.8255 | 0.618 | |

Ultimately, multi-dose clonogenic survival assays were performed, using 'multi-target single hitting' model fitting the survival curve. As figure 6*c,d* shows, compared with the control group, the cell survival curve is significantly decreased after NCs treatment, indicating that more cells are killed under the same radiation, which directly performed the radiation enhancement effect of the NCs. Furthermore, it also means that less radiation energy is required to achieve the same curative effect, when NCs is used simultaneously. That is, less damage is caused to surrounding normal tissues, which is of great significance in clinic research. To this end, dose enhancement factor (DEF) was calculated to evaluate the dose efficiency enhancement effect of NCs in dosiology [43,44], which is defined as the dose ratio of 90% SF between the control and NCs groups. As the figures exhibit, the DEF values are 1.423 for H1975 cells at 15 μg ml$^{-1}$ and 1.68 for A549 cells at 10 μg ml$^{-1}$. It means that, with the aid of NCs, the dose efficiency will be improved by nearly 1.5 times and the damage to surrounding normal tissues will be reduced by nearly 1.5 times. It describes the radiotherapy enhancement effect of NCs quantitatively.

Moreover, through function fitting, the parameters of $D_0$ and $N$ were obtained, as listed in table 1. They represent the mean lethal dose and the number of targets in DNA respectively in radiobiology. It can be seen that the value of $D_0$ reduces significantly after NCs treatment. It means that the mean lethal dose of cells cultured with NCs is reduced, implying the cell radiosensitivity improvement effect of the NCs. Therefore, radiation sensitivity enhancement ratio (SER) was used to evaluate the biological sensitivity enhancement of the NCs [4,16], which is defined as the $D_0$ ratio between the

control and NCs groups. As table 1 exhibits, it is 1.152 for H1975 cells at 15 µg ml$^{-1}$ and 1.998 for A549 cells at 10 µg ml$^{-1}$, proving the cell radiosensitivity improvement effect of the NCs in biological aspect. In addition. $D_q$ represents prospective domain dose. As the product of $D_0$ and $N$, it reflects the cell's ability for sublethal damage repair [4]. As can be seen in table 1, its value reduces as well, suggesting the ability of interfering the repair of DNA damage of the NCs. Therefore, the radiotherapy enhancement effect of the NCs has been exhibited and demonstrated through dosiology and radiobiology aspects, revealing the radiation and cell radiosensitivity enhancement of the NCs, which evidences the potential of the NCs to become good radiotherapy sensitizers in the future.

# 4. Discussion

In recent years, nanotechnology has been applied in various fields [1,3] and the research on novel materials fabrication has attracted increasing attention [25,26]. For instance, in 2017, He Wenya *et al.* [57] reported on plasmonic titanium nitride nanoparticles for photothermal cancer therapy. Min Yuanzeng *et al.* [58] developed antigen-capturing nanoparticles that can improve the abscopal effect in cancer immunotherapy. In the present work, a hypoxic drug carrier, GO-MI NCs, has been designed, in expectation of combating drug/radio-resistance of hypoxic tumours. Furthermore, the carrier was loaded with FePt MNPs and 5-FU chemotherapeutics [59,60], designated as MGO/FU-MI NCs, to simultaneously possess the ability of radiotherapy and chemotherapy efficiency improvement. In addition, MGO/FU and MGO-MI NCs were synthesized in the study as positive controls to clarify the antitumour effect of each drug loaded.

Primarily, as depicted in figure 1*b*, the antitumour effect of MGO/FU-MI NCs can be summarized as follows. Firstly, the drug carrier system has the recommended biological compatibility, which is directly reflected in the TEM images of cellular drug uptake. The microcytotoxicity of MGO/FU and MGO-MI NCs in MTT assay demonstrates this point indirectly as well, of which the cell proliferation inhibition effect is negligible even at a high concentration (100 µg ml$^{-1}$). Moreover, the loading cargos, 5-FU can interfere with DNA synthesis and repair of cells [49,50], which leads to S phase arresting in cell cycle analysis and $D_q$ values decrease (table 1). Furthermore, the FePt MNPs on the carrier can cause intracellular ROS augment through the Fenton reaction [13,56], which has been proved by intracellular ROS detection. Additionally, the hypoxia-response ingredient, MI leads to the hypoxia sensitivity of the NCs which has testified by hypoxic MTT assay. It is also connected with the cell radiosensitivity improvement, which could augment the fraction of G2 phase in cell cycle. Thus, the antitumour mechanism of each component has been revealed so far. However, there exist two principal themes throughout the study that need further discussion. One is the additive effect between MI and 5-FU, and the other is the cell-type dependent cytotoxicity phenomena, hypoxia sensitivity in particular.

The additive effect between MI and 5-FU appeared throughout the *in vitro* experiments, including MTT, cell cycle and ROS analyses. First of all, in the MTT assay, the inhibitory effect of 2 µg ml$^{-1}$ MGO/FU-MI NCs is approximately equivalent to that of 100 µg ml$^{-1}$ the other two NCs, which is nearly 50-fold stronger. Longley *et al.* [48] have reported the improved response rates when 5-FU is combined with other chemotherapeutics but MI is not mentioned. Thus, wild speculation is made that the antitumour rates could be improved when 5-FU and MI are used together. The additive effect between 5-FU and MI was firstly demonstrated by cell cycle analysis. In the MGO/FU-MI treatment group, the fraction of S and G2 phase are both augmented and nearly to the same degree as the other two NCs treatment groups, proving that MGO/FU-MI NCs stimulates the same cellular regulatory mechanisms of both MI and 5-FU. It means that the additive effect between 5-FU and MI is an additional phenomenon which does not suppress the effect of any drugs. Moreover, in the intracellular ROS detection, the intracellular ROS level is more than onefold augmented with MGO/FU-MI NCs treatment, implying there exist other interactions of the two drugs that might stimulate the excessive production of ROS. It indicates that the additive effect is not simply the sum of the effects of the two drugs, but rather that there are interactions between the two drugs. Further investigation is required before it can be revealed.

Besides, the antitumour effect of any NCs is cytotype dependent, showing a tender influence on A549 cells in all analyses. The phenomenon highlights the fact that all cytotoxicity outcomes are based on cellular uptake and metabolic mechanisms, which is a comprehensive result of drug effect and cellular regulatory mechanism. Take the well-known phenomenon of drug-resistance in hypoxic cells as an example. Researchers have found that the tumour cells have to change their intracellular molecules and regulatory pathways to adapt to the hypoxia environment [61]. In this process, the hypoxia

inducible factors-1α (HIFs-1α) is a regulator that is involved in many intracellular signalling pathways that are related to the chemoresistance in the hypoxic cells [62]. Moreover, the hypoxia condition can also regulate the expression of proteins, such as BCL-2, Livin [63], which can change the metabolic state of cells to influence their drug resistance. Therefore, the drug resistance caused by hypoxia condition is also cytotype dependent, which is related to the dominant regulatory pathway in different cell types [46]. Thus, the cytotoxicity of the NCs is similar, which is related to the cytotype and environment dependent cellular uptake and metabolic mechanisms as well.

Upon these points mentioned above, the results of hypoxia sensitivity tests in the study are understandable. As is exhibited (figure 4g), the decreased cell viability of the two types of cells treated with MGO-MI NCs under hypoxia condition is evidence of the hypoxia sensitivity of MI. However, the cell viability of H1975 cells treated with MGO/FU NCs under hypoxia condition is increased compared to normoxia condition, while the result of A549 cells is contrary. The difference indicates that the hypoxic environment activates diverse cellular regulatory mechanisms upon the two types of cells [46]. It leads to drug-resistance of H1975 cells while A549 does not. Thus, we know that the decreased cell viability of A549 cells treated with MGO/FU-MI NCs is not evidence of hypoxia sensitivity of the NCs. Furthermore, comparing with the result of H1975 cells treated with MGO-MI NCs, the difference may be caused by the adding of 5-FU as well. In another words, the interactions between 5-FU and MI might stir up some intracellular signalling pathways that affect the hypoxia-response of MI and lead to the failure or reduction of hypoxia sensitivity of the NCs upon H1975 cells. However, all speculations require further pharmacokinetic investigations.

Ultimately, it is necessary to mention the lethal mechanism of radiotherapy to comprehend the radiotherapy improvement mechanism of the NCs. On radiobiology, the effects of ionizing radiation on living organisms could be divided into three stages, which are physical, chemical and biological. As the X-ray passed through the body, secondary electrons are generated along the ray path, which ionize water molecules and lose energy [6]. This process lasts only $10^{-12}$ s and is called the physical stage, during which the radical anions are produced [23,64]. The radical anions cause single strand breaks (SSBs) and double strand breaks (DSBs) of DNA, and molecular structure damage of cells, which is called the chemical stage, lasting $10^{-5}$ s [13]. Then the cells would repair the damage through a series of enzyme reactions, and the damage that can be repaired is called sublethal damage, like SSBs. Cells unable to be repaired, like DSBs, would lead to mitotic arrest and cell death [65]. This process would last seconds or happen decades later, which is called the biological stage. Therefore, in the design of the NCs, FePt MNPs will act as high-Z type radiosensitizers [6,44], which enhance the radiation energy deposition within a tumour through its high radiation rays absorption coefficient [13,36] during the physical stage. Moreover, MI can assist the action of radical anions through simulating the action of oxygen [9,64] in the chemical stage. Whereas the physical and chemical stages are short, the magnitude of the enhancement effect is dependent on the amount of NCs in cells. It has been proven by the releasing curve of the NCs that it is very stable in solutions. Therefore, with the uptake of cells, the NCs will have a good radiation enhancement effect during the physical and chemical lethal stage of radiotherapy. Furthermore, due to a long incubation time in biological stage, the 5-FU will give full play to its role in hindering the repair sublethal damage through its function of DNA synthesis interference [34,66]. In summary, the three ingredients of NCs can greatly enhance the radiotherapy efficiency in all three stage of the lethal process of radiotherapy, including physical, chemical and biologic stage. It constitutes a self-amplified radiation sensitization system. Therefore, not only the radiation efficiency but also the cell radiosensitivity is enhanced after the NCs treatment, presenting a potential radiosensitizer candidate for radiotherapy. In addition, it should be emphasized that longer incubation time is required to make the NCs make the most of their role in radiotherapy improvement.

Overall, the present study recommends an intelligent nanoplatform which attracts attention by its conspicuous properties of hypoxia sensitivity and chemoradiotherapy co-enhancement effect. Tremendous enhancement effects of cytotoxicity, radiation and cell radiosensitivity of the NCs are performed and demonstrated in the study. Although the findings are encouraging, there is still much to be done to perfect the process. Firstly and constitutionally, the present drug delivery system should be further modified by biological target to expand the scope of its antitumour action. In conventional chemotherapy, chemotherapeutic drugs are to go around the whole body via intravenous injection to attack cancer cells, including metastatic tumours. The drugs themselves are not targeted, and their enrichment in the tumour sites is mainly due to the ubiquitous enhanced permeability and retention (EPR) effect in solid tumours [25,67]. However, the EPR effect always applies to macromolecular substances. Therefore, when it comes to small molecule chemotherapeutic drugs like 5-FU, they are not targeted at all, and they will attack all cells in the body, including cancer cells, metastatic tumours

and normal tissues. As a result, the problem of systemic toxicity arises [7]. As a consequence, the hypoxic drug delivery system was designed and synthesized in the study to solve the problem of large dosage and low efficiency of clinical medication. However, every coin has two sides, the present design is imperfect, which will limit its antitumour effect on metastatic tumours that may exist anywhere within the body. Therefore, further modification is required to expand its scope of antitumour effect. Secondly, further optimization of the consumption of some of the raw materials in the present synthetic method is necessary to perfect the loading characteristics of the NCs. Thirdly, the hypoxia sensitivity of the NCs is tested just under physical hypoxia which is caused by mixing of different fractions of gases. The conditions of chemical and biological hypoxia which is caused by chemical and biological oxygen consumption should be used to test the hypoxia sensitivity of the NCs thoroughly. Fourthly, more quantitative analyses are needed to bring the novel materials closer to further application, such as optimal loading amount and the accurate drug pharmacokinetics in tumour volumes. Fifthly, the problems related to cell-type dependence and pharmacokinetics in the present study require further investigation to comprehend the antitumour effect of the NCs completely. Sixthly, *in vivo* studies are required in the present research to examine the *in vivo* properties of NCs. Thus, more optimization and investigations should be carried out to perfect the present study and explore a more complete biological mechanism of MGO/FU-MI NCs.

## 5. Conclusion

In conclusion, MGO/FU-MI NCs are designed and synthesized to serve as nanosensitizers to achieve the co-enhancement of chemoradiotherapy in the study. During *in vitro* experiments, the NCs exhibit an inspiring additive effect among MI, 5-FU and FePt MNPs, which cause intracellular ROS burst and significant cell proliferation suppression effect, greatly improving chemotherapy efficiency. In addition, the self-amplified radiation sensitization system composed of the three ingredients of the NCs tremendously enhance the radiotherapy efficiency, in which FePt MNPs cause a high radiation energy deposition while the MI lead to radical anions overproduction, guaranteeing the radiotherapy enhancement in physical and chemical aspects. Furthermore, 5-FU hinders the sublethal damage repair of DNA which secures the radical damage in biological aspect. Thus, the NCs achieve a self-amplified chemoradiotherapy co-enhancement. Although the results are encouraging, future investigations on quantitative analyses and *in vivo* application should be carried out to complete the biological antitumour mechanism *in vivo*. Thus, the study reveals a novel and potential nanoplatform for tumour treatment which could improve the synergistic effect of chemoradiotherapy.

Data accessibility. Our data, which include physical, biological detection and releasing characteristics deposits, can be found at Dryad: https://doi.org/10.5061/dryad.n0n3k6b.

Authors' contributions. C.Y. and S.P. synthesized the materials and carried out the *in vitro* experiments, participated in data analysis, article writing and figures charting. Y.S. and H.M. in charge of the materials design, assisted the materials synthesis. M.L. polished the critical content of the article. S.M. and Y.L. perfected the design of the study. R.X., C.X. and H.Q. proposed the design of new materials and finally approved the version. All authors reviewed the manuscript.

Competing interests. There are no conflicts to declare.

Funding. Financial support mainly came from Natural Science Foundation of China (grant nos. 81572967 and 81372498).

Acknowledgement. We thank the national financially support of Natural Science Foundation of China (no. 81572967 and no. 81372498) and National Key clinical speciality construction program of China (no. [2013]544). The instructional support of Zhongnan Hospital of Wuhan University, Technology and Innovation Seed Fund (no. znpy2016050 and no. znpy2017049). We are also grateful to the assistance of Huanghe Talents Plan and Medical Physics Teaching and research Fund of Elekta & Wuhan University (no. 250000200). We also thank for the assistance from Yaxi Yang, Jian He, Pengyuan Qi and others who makes contributions to the study.

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
