## [Reviewer comments · Royal Society Open Science]

Review History

RSOS-181790.R0 (Original submission)

Review form: Reviewer 1

Is the manuscript scientifically sound in its present form?

No

Are the interpretations and conclusions justified by the results?

No

Is the language acceptable?

No

Is it clear how to access all supporting data?

No

Do you have any ethical concerns with this paper?

No

Have you any concerns about statistical analyses in this paper?

No

Recommendation?

Major revision is needed (please make suggestions in comments)

Comments to the Author(s)

This research which is based on synthesising a composite nanoparticle including chemotherapy agent - 5FU is interesting and worth considering for publication. However, there are some important questions which needs to be answered before such consideration. And the paper is poorly written it needs major re-writing especially discussion section.

The main question is about the fact that if we were to target 5FU as suggested by this paper then its role to attack metastases will be removed. Can the authors address this point using references please. Also I have listed some other questions and suggestions for improving the manuscript in the attached file (Appendix A).

Review form: Reviewer 2

Is the manuscript scientifically sound in its present form?

Yes

Are the interpretations and conclusions justified by the results?

Yes

Is the language acceptable?

Yes

Is it clear how to access all supporting data?

Not Applicable

Do you have any ethical concerns with this paper?

No

Have you any concerns about statistical analyses in this paper?

No

Recommendation?

Accept with minor revision (please list in comments)

Comments to the Author(s)

In this paper, the authors reported a promising novel multifunctional platform which displayed a self-amplified effect that activated chemoradiotherapy co-enhancement. The research is interesting. And there are some suggestions to improve this article.

1) I think the words "panting efficiency (PE)" (Page 5, line 41) should be changed to "plating efficiency (PE)" and the words "the sensitization enhancement radio (SER)" (Page 13, line 8-9) should be changed to "the sensitization enhancement ratio (SER). Please check carefully.

- 2) What is the meaning of the word "SF2" in the manuscript? Maybe it should be changed to "SF" (survival fraction).
- 3) From the Figure 6a, we know the value of S of H1975 cells treated by MGO/FU is 28.7 and that of S of H1975 cells treated by MGO/FU-MI NCs is 27.8. But in Figure 6b, we observe the values represented by the graph are different which the value of S of H1975 cells treated by MGO/FU-MI NCs is larger than that of S of H1975 cells treated by MGO/FU. Maybe the authors should pay attention to this.
- 4) Please check the manuscript carefully.

Review form: Reviewer 3

Is the manuscript scientifically sound in its present form?

Yes

Are the interpretations and conclusions justified by the results?

Yes

Is the language acceptable?

Yes

Is it clear how to access all supporting data?

Yes

Do you have any ethical concerns with this paper?

No

Have you any concerns about statistical analyses in this paper?

No

Recommendation?

Accept with minor revision (please list in comments)

Comments to the Author(s)

In this manuscript, the authors discussed the development of a multifunctional graphene oxide based nanocomposites to deliver metronidazole, 5-fluorouracil, and PePt magnetic nanoparticles for chemoradiotherapy. The integration of hypoxic drug carrier, radiosensitizer and chemotherapeutic drugs confers hypoxia-sensitivity and chemoradiotherapy co-enhancement functions to a single nano-platform. The detailed in vitro studies of the nanocomposites using A549 and H1954 cells exhibit significant cell proliferation suppression, intracellular ROS burst, and improved radiosensitivity, which could act as a potential multifunctional nano-platform for tumor treatment to improve the synergistic effect of chemoradiotherapy. However, some important parameters of this nano-platform that could affect the in vivo application and translation for nano-drug delivery systems are missing. There are some major concerns associated with the current version of submission which are listed below.

Comments:

1. The PePt MNPs and 5-FU were loaded on the MGO/FU-MI NCs via physisorption. One concern with the physical loading systems is the cargo loading stability and release kinetics. It is known that the cargoes may release during circulation, which leads to low drug delivery to the tumors, poor treatment efficiency and potentially toxicity to normal tissues. It is important for the authors to add the loading stability of the NCs and the release profile in both physiological and

tumor environment condition.

2. For any drug delivery system, the loading efficiency and loading capacities are critical for effective tumor therapy. These data are missing for the MGO/MI, MGO/FU, and MGO/FU-MI nanocomposites. Additionally, it is useful for the audience to understand the advantages of this system by showing the data regarding the optimal cargo loading amount and ratio since these data may affect the cell viability and ROS production results.

3. It is confusing that the cancer cells showed reduced cell viability under normoxia condition in the presence of MGO/FU-MI NCs in Figure 4g. Typically, the hypoxia sensitivity of NCs should facilitate cell proliferation suppression under hypoxia condition.

4. In page 12 line 4: "The composites loading 5-FU showed an S phase arresting, the fraction of which was nearly one-times augment for H1975 cells cultured with MGO/FU NCs." Is it a typo that the "MGO/FU" should be "control"? As the composites loading 5-FU should also be MGO/FU NCs. And, "one-times" should be "one-time".

5. In figure 2f, what are the two peaks at 9 keV and 11.4 keV attributed to?

Decision letter (RSOS-181790.R0)

21-Feb-2019

Dear Miss Yang,

The editors assigned to your paper ("Development of a hypoxic nanocomposite containing high-Z element as 5-Fluorouracil carrier activated self-amplified chemoradiotherapy co-enhancement") have now received comments from reviewers. We would like you to revise your paper in accordance with the referee and Associate Editor suggestions which can be found below (not including confidential reports to the Editor). Please note this decision does not guarantee eventual acceptance.

Please submit a copy of your revised paper before 16-Mar-2019. Please note that the revision deadline will expire at 00.00am on this date. If we do not hear from you within this time then it will be assumed that the paper has been withdrawn. In exceptional circumstances, extensions may be possible if agreed with the Editorial Office in advance. We do not allow multiple rounds of revision so we urge you to make every effort to fully address all of the comments at this stage. If deemed necessary by the Editors, your manuscript will be sent back to one or more of the original reviewers for assessment. If the original reviewers are not available, we may invite new reviewers.

- Data accessibility

If you wish to submit your supporting data or code to Dryad (<http://datadryad.org/>), or modify your current submission to dryad, please use the following link:
<http://datadryad.org/submit?journalID=RSOS&manu=RSOS-181790>

- Competing interests

- Authors' contributions

- Acknowledgements

- Funding statement

on behalf of Dr Derek Abbott (Associate Editor) and R. Kerry Rowe (Subject Editor)
openscience@royalsociety.org

Comments to Author:

Reviewers' Comments to Author:
Reviewer: 1

Comments to the Author(s)

This research which is based on synthesising a composite nanoparticle including chemotherapy agent - 5FU is interesting and worth considering for publication. However, there are some important questions which needs to be answered before such consideration. And the paper is poorly written it needs major re-writing especially discussion section.

The main question is about the fact that if we were to target 5FU as suggested by this paper then its role to attack metastases will be removed. Can the authors address this point using references please. Also I have listed some other questions and suggestions for improving the manuscript in the attached file

Reviewer: 2

Comments to the Author(s)

In this paper, the authors reported a promising novel multifunctional platform which displayed a self-amplified effect that activated chemoradiotherapy co-enhancement. The research is interesting. And there are some suggestions to improve this article.

- 1) I think the words "panting efficiency (PE)" (Page 5, line 41) should be changed to "plating efficiency (PE)" and the words "the sensitization enhancement radio (SER)" (Page 13, line 8-9) should be changed to "the sensitization enhancement ratio (SER). Please check carefully.
- 2) What is the meaning of the word "SF2" in the manuscript? Maybe it should be changed to "SF" (survival fraction).
- 3) From the Figure 6a, we know the value of S of H1975 cells treated by MGO/FU is 28.7 and that of S of H1975 cells treated by MGO/FU-MI NCs is 27.8. But in Figure 6b, we observe the values represented by the graph are different which the value of S of H1975 cells treated by MGO/FU-MI NCs is larger than that of S of H1975 cells treated by MGO/FU. Maybe the authors should pay attention to this.
- 4) Please check the manuscript carefully.

Reviewer: 3

Comments to the Author(s)

In this manuscript, the authors discussed the development of a multifunctional graphene oxide based nanocomposites to deliver metronidazole, 5-fluorouracil, and PePt magnetic nanoparticles

for chemoradiotherapy. The integration of hypoxic drug carrier, radiosensitizer and chemotherapeutic drugs confers hypoxia-sensitivity and chemoradiotherapy co-enhancement functions to a single nano-platform. The detailed in vitro studies of the nanocomposites using A549 and H1954 cells exhibit significant cell proliferation suppression, intracellular ROS burst, and improved radiosensitivity, which could act as a potential multifunctional nano-platform for tumor treatment to improve the synergistic effect of chemoradiotherapy. However, some important parameters of this nano-platform that could affect the in vivo application and translation for nano-drug delivery systems are missing. There are some major concerns associated with the current version of submission which are listed below.

Comments:

1. The PePt MNPs and 5-FU were loaded on the MGO/FU-MI NCs via physisorption. One concern with the physical loading systems is the cargo loading stability and release kinetics. It is known that the cargoes may release during circulation, which leads to low drug delivery to the tumors, poor treatment efficiency and potentially toxicity to normal tissues. It is important for the authors to add the loading stability of the NCs and the release profile in both physiological and tumor environment condition.
2. For any drug delivery system, the loading efficiency and loading capacities are critical for effective tumor therapy. These data are missing for the MGO/MI, MGO/FU, and MGO/FU-MI nanocomposites. Additionally, it is useful for the audience to understand the advantages of this system by showing the data regarding the optimal cargo loading amount and ratio since these data may affect the cell viability and ROS production results.
3. It is confusing that the cancer cells showed reduced cell viability under normoxia condition in the presence of MGO/FU-MI NCs in Figure 4g. Typically, the hypoxia sensitivity of NCs should facilitate cell proliferation suppression under hypoxia condition.
4. In page 12 line 4: "The composites loading 5-FU showed an S phase arresting, the fraction of which was nearly one-times augment for H1975 cells cultured with MGO/FU NCs." Is it a typo that the "MGO/FU" should be "control"? As the composites loading 5-FU should also be MGO/FU NCs. And, "one-times" should be "one-time".
5. In figure 2f, what are the two peaks at 9 keV and 11.4 keV attributed to?

Author's Response to Decision Letter for (RSOS-181790.R0)

See Appendix B.

RSOS-181790.R1 (Revision)

Review form: Reviewer 1

Is the manuscript scientifically sound in its present form?

Yes

Are the interpretations and conclusions justified by the results?

No

Is the language acceptable?

Yes

Is it clear how to access all supporting data?

Yes

Do you have any ethical concerns with this paper?

No

Have you any concerns about statistical analyses in this paper?

No

Recommendation?

Major revision is needed (please make suggestions in comments)

Comments to the Author(s)

All aspects of this manuscript are cleared now. Except one point: the 5FU when used in cancer patients it is not only to affect the cancer volume but also if any metastasises that might exist anywhere within the body. Hence, targeting 5FU to the tumour volume will result into limitation of its applications. This point needs to be made clear by the authors for the readers. They attempted in their revision of the text but it is still not clear enough. I think the authors didn't understand the request and I hope now it is clear. So please address this issue before the publication of the paper

Review form: Reviewer 2

Is the manuscript scientifically sound in its present form?

Yes

Are the interpretations and conclusions justified by the results?

Yes

Is the language acceptable?

Yes

Is it clear how to access all supporting data?

No

Do you have any ethical concerns with this paper?

No

Have you any concerns about statistical analyses in this paper?

No

Recommendation?

Accept as is

Comments to the Author(s)

The authors have further improved the manuscript. I think it can be accepted for publication now.

Review form: Reviewer 3

Is the manuscript scientifically sound in its present form?

Yes

Are the interpretations and conclusions justified by the results?

Yes

Is the language acceptable?

No

Is it clear how to access all supporting data?

Yes

Do you have any ethical concerns with this paper?

No

Have you any concerns about statistical analyses in this paper?

No

Recommendation?

Major revision is needed (please make suggestions in comments)

Comments to the Author(s)

The authors added more details to the resubmitted version according to the comments. However, this version still have some issues that the authors need to address.

1) I didn't clarify the physiological and tumor environment conditions, which are not water and PBS. I would like the author to compare the stability and release of the NPs at different pH, normally, 7 and 5-6. (doi: 10.1186/1475-2867-13-89). In addition, it will be more comprehensive to use cumulative release curve in Figure 6.

2) The authors added the loading efficiency of the NPs in the present synthetic method. However, I would prefer to know the loading capacity, which could represent the maximal loading dose of cargos on NPs, while the loading efficiency depends on the feeding amount.

Loading capacity (%) = (the weight of loaded drug/ the weight of NPs)

3) The author gave an explanation about the cell-type dependence phenomena, please add some reference support. However, the writing of the corresponding paragraph is quite confusing. For example, what is the meaning of the sentence "The hypoxia sensitivity tests is vary an example of that"? Not only this paragraph, all the discussion section needs to be carefully revised. There are even many typos in the rebuttal letter.

4) 5) no further comments.

Decision letter (RSOS-181790.R1)

08-Apr-2019

Dear Miss Yang:

Manuscript ID RSOS-181790.R1 entitled "Development of a hypoxic nanocomposite containing high-Z element as 5-Fluorouracil carrier activated self-amplified chemoradiotherapy co-

enhancement" which you submitted to Royal Society Open Science, has been reviewed. The comments of the reviewer(s) are included at the bottom of this letter.

Please submit a copy of your revised paper before 01-May-2019. Please note that the revision deadline will expire at 00.00am on this date. If we do not hear from you within this time then it will be assumed that the paper has been withdrawn. In exceptional circumstances, extensions may be possible if agreed with the Editorial Office in advance. We do not allow multiple rounds of revision so we urge you to make every effort to fully address all of the comments at this stage. If deemed necessary by the Editors, your manuscript will be sent back to one or more of the original reviewers for assessment. If the original reviewers are not available we may invite new reviewers.

- Ethics statement

- Data accessibility

- Competing interests

- Authors' contributions

All submissions, other than those with a single author, must include an Authors' Contributions section which individually lists the specific contribution of each author. The list of Authors should meet all of the following criteria; 1) substantial contributions to conception and design, or

acquisition of data, or analysis and interpretation of data; 2) drafting the article or revising it critically for important intellectual content; and 3) final approval of the version to be published.

- Acknowledgements

- Funding statement

on behalf of Dr Derek Abbott (Associate Editor) and Professor R. Kerry Rowe (Subject Editor)
openscience@royalsociety.org

Reviewer comments to Author:

Reviewer: 3

Comments to the Author(s)

The authors added more details to the resubmitted version according to the comments. However, this version still have some issues that the authors need to address.

1) I didn't clarify the physiological and tumor environment conditions, which are not water and PBS. I would like the author to compare the stability and release of the NPs at different pH, normally, 7 and 5-6. (doi: 10.1186/1475-2867-13-89). In addition, it will be more comprehensive to use cumulative release curve in Figure 6.

2) The authors added the loading efficiency of the NPs in the present synthetic method. However, I would prefer to know the loading capacity, which could represent the maximal loading dose of cargos on NPs, while the loading efficiency depends on the feeding amount.

Loading capacity (%) = (the weight of loaded drug/ the weight of NPs)

3) The author gave an explanation about the cell-type dependence phenomena, please add some reference support. However, the writing of the corresponding paragraph is quite confusing. For example, what is the meaning of the sentence "The hypoxia sensitivity tests is vary an example of that"? Not only this paragraph, all the discussion section needs to be carefully revised. There are even many typos in the rebuttal letter.

4) 5) no further comments.

Reviewer: 2

Comments to the Author(s)

The authors have further improved the manuscript. I think it can be accepted for publication now.

Reviewer: 1

Comments to the Author(s)

All aspects of this manuscript are cleared now. Except one point: the 5FU when used in cancer patients it is not only to affect the cancer volume but also if any metastasises that might exist anywhere within the body. Hence, targeting 5FU to the tumour volume will result into limitation of its applications. This point needs to be made clear by the authors for the readers. They attempted in their revision of the text but it is still not clear enough. I think the authors didn't understand the request and I hope now it is clear. So please address this issue before the publication of the paper

Author's Response to Decision Letter for (RSOS-181790.R1)

See Appendix C.

RSOS-181790.R2 (Revision)

Review form: Reviewer 1

Is the manuscript scientifically sound in its present form?

Yes

Are the interpretations and conclusions justified by the results?

Yes

Is the language acceptable?

Yes

Is it clear how to access all supporting data?

Not Applicable

Do you have any ethical concerns with this paper?

No

Have you any concerns about statistical analyses in this paper?

I do not feel qualified to assess the statistics

Recommendation?

Accept as is

Comments to the Author(s)

I am somehow now satisfied with the discussion you presented regarding losing some benefits of 5FU in the case of targeting it

Review form: Reviewer 3

Is the manuscript scientifically sound in its present form?

Yes

Are the interpretations and conclusions justified by the results?

Yes

Is the language acceptable?

Yes

Is it clear how to access all supporting data?

Yes

Do you have any ethical concerns with this paper?

No

Have you any concerns about statistical analyses in this paper?

No

Recommendation?

Accept as is

Comments to the Author(s)

I found 1 typo on line 31 page 13: "knew" should be "known". Except that, the manuscript has been greatly improved. I suggest it can be accepted for publication now.

Decision letter (RSOS-181790.R2)

23-May-2019

Dear Miss Yang,

I am pleased to inform you that your manuscript entitled "Development of a hypoxic nanocomposite containing high-Z element as 5-Fluorouracil carrier activated self-amplified chemoradiotherapy co-enhancement" is now accepted for publication in Royal Society Open Science.

You can expect to receive a proof of your article in the near future. Please contact the editorial

office (openscience_proofs@royalsociety.org and openscience@royalsociety.org) to let us know if you are likely to be away from e-mail contact. Due to rapid publication and an extremely tight schedule, if comments are not received, your paper may experience a delay in publication.

on behalf of Dr Derek Abbott (Associate Editor) and R. Kerry Rowe (Subject Editor)
openscience@royalsociety.org

Reviewer comments to Author:
Reviewer: 3

Comments to the Author(s)
I found 1 typo on line 31 page 13: "knew" should be "known". Except that, the manuscript has been greatly improved. I suggest it can be accepted for publication now.

Reviewer: 1

Comments to the Author(s)
I am somehow now satisfied with the discussion you presented regarding loosing some benefits of 5FU in the case of targeting it

Appendix A

RoyalSociety-18

General;

1. The 5FU is meant to go around whole body and affect the DNAs of cancer cells including metastasis. If it is i.e. the 5-FU targeted, then it will miss its effects on remote cancer cells.
2. Any results to be shown in the abstract??
3. There is no word as "Leaded" please replace it through out the manuscript into "Lead" or "Led"
4. The method for measuring the levels of generated ROS as briefly mentioned in section 2.6.2 is not clear and not explained to the reader to understand how are the ROSs detected and counted??

Corrections;

ABSTRACT:

1. Line 4; "drug delivery system with 246nm" what does this mean? Is this diameter? If so of what?
2. Line 7 "Chemo-therapeutics " remove 's" at the end

INTRODUCTION

1. Page 2- line 7- "attention that fabricating" replace "that" by "to"
2. Second paragraph: line 6; "to delivery insulin for .." change "delivery" to "Deliver"
3. Same line as above: "where"? does not make sense. Change the whole sentence o may be replace 'where" with "were"?
4. Third paragraph; second sentence: Change "In especial" to "Especially"
5. Figures 2-a and 2-b are not clear and the scale is unreadable

METHOS:

1. Section-2.2.1 second sentence is not clear needs re-writing
2. Section 2.2.2 Replace "500" by a word such as "The" etc...
3. Section 2.3 "Dynamic light scatter" change to "Dynamic Light Scattering device"
4. Section 2.6.2 Explain what is meant by "DCF"

RESULTS:

1. Section 3.1 second sentence: remove "At the meantime" and replace it with "The"
2. Same sentence; Following FePt MNPs replace "was" with "were"
3. Section 3.1 line 9 : change "exhibited" to "exhibits"
4. Similarly few other words need to be changed for ending into "ed" into "s"
5. Section 3.2 line 4: change 'cargos loaded' to "loaded cargos"
6. Page 10: Font sizes in some figure 4 sections are not readable.
7. Also in page 10. The third line from bottom of the page; "replace "responsive" with "response"

8. There is no explanation for the results of Hypoxia response differences between the 2 type of cells??
9. The second sentence on page 11 is not clear. Please re-write it. Something as "...Hypoxia-response was cell difference" does not make sense. Do you mean that Hypoxia depends on cell type?
10. Section 3.3: First sentence; add "for" before 24h
11. Same sentence; add "concentration of" before 20
12. Also this sentence claims that ROS is measured. But How? At least refer to the method. How reliable it is etc..
13. The sentence in the middle of the second paragraph page 12: "As depicted in figure 5d,..." There is no explanation for this observation????
14. The following sentence mentions "burst of ROS" How? And why?
15. This section 3.3 is very shallow it needs expansion and further explanation & discussion of the results
16. Section 3.4 is very poorly written. It is not the matter of language only but structure and material. Many sentences don't make sense at all. For instance the second one which starts with "As expected.." as expected based on what??. I tried to help organising the section but I realised that I will end up re-writing it which is not my responsibility.
17. Figure 6-b should be marked 2Gy to compare it with a
18. Figure 6-d; how is this curve fitting done? Can it be shown what degree of polynomial is used? As it does not look right
19. DEF values listed in figure 6 are not defined and not explained how are they obtained?
20. Two terms DEF and SER are used through out the paper it needs some explanation as to what is the difference between the two terms if any
21. Section 4 "Discussion" is poorly written, ambiguous, and about the results but about the idea of the project. Discussion should be about the results and what they mean and what are their limitations etc..
22. The conclusion "section 5" also will benefit greatly from re-writing. Example; The sentence before the last: "future investigation was needed" this sentence doesn't make any sense. If it is future how is it "was"??

Appendix B

Dear Editor and Reviewers:

Thank you for your kind letter of decision and reviewers' comments on our previous manuscript, RSOS-181790 ("Development of a hypoxic nanocomposite containing high-Z element as 5-Fluorouracil carrier activated self-amplified chemoradiotherapy co-enhancement"), on February 21st, 2019. Those comments are all valuable and very helpful for revising and improving our manuscript. We have studied the comments carefully and revised the manuscript accordingly, we have also carefully checked it to minimize spelling and grammar mistakes, which we hope meet with approval. The following is our description on revision, which is divided into three parts. Among them,

Part 1: List of Actions, which is a summary statement that covering all changes we have made in the revision.

Part 2: Authors' Response to Reviewers, which is our response to the comments from reviewers.

Part 3: Answers to Questions Listed in Attachment form Reviewer 1, which is the specific answers to the questions mentioned by Reviewer 1 in attachment.

Part 1: List of Actions

LOA 1: We have changed the font of the title and subtitles to make it clearer.

LOA 2: We have made a small change to the authors. The contributors who haven't meet the authorship criteria are removed and included in the acknowledgements section as suggested. Therefore, the section of author contributions and

acknowledgements are changed corresponding.

LOA 3: We have extensively revised the previous manuscript as suggested by reviewer 1, including abstract, introduction, experimental section, results, discussion, conclusion.

LOA 4: We have changed some of the words and phrases in abstract section as suggested by reviewer 1 to make it clearer and more correct, such as “with an average size of 243nm (line 4)” and “chemotherapeutic drugs (line 7)”. In addition, we have totally rewritten the expression of results shown in abstract (line 9-13), to stress the key points of our study. (abstract section, page 1)

LOA 5: We have changed some of the words, phrases and sentences of the Introduction as suggested by reviewer 1, such as “deliver insulin (line 6, paragraph 2, page 2)”, “Especially (line 2, paragraph 3, page 2)” and “Ji-Chang Yu et al. developed a hypoxia-sensitive vesicle to deliver insulin for diabetes treatment, using NI as hypoxic ingredient likewise (line 6-7, paragraph 2, page 2)”. Moreover, we have totally rewritten the first and last paragraph (page 1-2) to make it more smooth and stress the important conclusion in our study.

LOA 6: We have rewritten some of the expression in section 2.2.1 as suggested by reviewer 1 to make the synthesis clearer (line 2-4, section 2.2.1, page 3).

LOA 7: We have supplemented the methods (section 2.4, Page 4), results (Figure 3d&e) and descriptions (paragraph 4, section 3.1, page 7-8) of the loading characters of NCs as suggested by reviewer 3.

LOA 8: We have supplemented the principles, methods, and procedures of the analysis of ROS in section 2.7.2 (page 5) as suggested by reviewer 1. Moreover, we have rewritten the descriptions (line 4-12, paragraph 2, page 9) and discussion (line 4-8, paragraph 1; line 13-15, paragraph 2, page 11) of ROS generation to make the results clearer and more comprehensible to audiences.

LOA 9: We have totally rewritten the section of 2.8 (page 5-6) and supplemented the equation of “multi target-single hitting” model (line 3, paragraph 2, section 2.8.3, page 6), the meaning of D_0 and N in radiobiology (line 4-5, paragraph 2, section 2.8.3, page 6) and the definition of SF_2 (last sentence of section 2.8.2), DEF , SER (paragraph 2, section 2.8.3, page 6) as suggested by reviewer 1.

LOA 10: We have completely changed the tense of results description to simple present tense as suggested by reviewer 1. And we have almost rewritten the descriptions of results, including the cytotoxicity (paragraph 2, section 3.2, page 8), hypoxia sensitivity (paragraph 3, section 3.2, page 8), cell cycle analysis (paragraph 1, section 3.3, page 9), ROS generation (paragraph 2, section 3.3, page 9) and the radiotherapy enhancement (section 3.4, page 9-10) to make them clearer and comprehensible.

LOA 11: We have totally rewritten the discussion section (page 10-12) as suggested by reviewer 1 to make it better organized. We have detailly discussed the experiment results (paragraph 2), the additive effect between the two drugs (paragraph 3), the cell-type dependence phenomena (paragraph 4) which includes the discussion of

hypoxia sensitivity (line 3-13, paragraph 4), the self-amplified radiation sensitization system (paragraph 5) and the limitations of the present study (paragraph 6)

LOA 12: We have almost rewritten the section 5 (page 12) as suggested by reviewer 1 to make it more general and covering the main points of the study.

LOA 13: We have supplemented some reference during the revision of manuscript.

LOA 14: We have changed the format of manuscript, removing the figures and supplementing the captions. Moreover, we have modified the font in figures and the captions accordingly to make it more readable as suggested by reviewers.

LOA 15: We have changed the spelling mistakes as suggested. Such as “planting efficiency (PE)” (line 9, section 2.8.1, page 5), “one-time” (line 4, page 9) and so on. Also we have changed some incorrect expression, such as change “the fraction of which was nearly one-time augment for H1975 cells cultured with MFO/FU NCs.” to “Compared with the control group, MGO/FU NCs shows an S phase arresting effect, the fraction of which is almost one-time enhanced on H1975 cells” (line 3-4, page 9). As suggested by reviewers.

LOA 16: We have changed the verb tense to simple present tense when describing result and conclusions, as suggested by reviewer 1.

Part 2: Authors' Responses to Reviewers:

Reviewer: 1

The reviewer's comments:

This research which is based on synthesising a composite nanoparticle including chemotherapy agent - 5FU is interesting and worth considering for publication. However, there are some important questions which needs to be answered before such consideration. And the paper is poorly written it needs major re-writing especially discussion section.

The main question is about the fact that if we were to target 5FU as suggested by this paper then its role to attack metastases will be removed. Can the authors address this point using references please. Also I have listed some other questions and suggestions for improving the manuscript in the attached file

The authors' responses:

First of all, we want to thank Reviewer: 1 for writing that **"This research ... is interesting and worth considering for publication."** Thank you for your recognition and feedback on our manuscript. Secondly, we would like to thank you again for your comments, corrections and suggestions on my pervious manuscript, which guide me to revise it, making it clearer, better organized, more correct and comprehensible for audiences. Specifically, we have partially rewritten the abstract (**LOA 4**) and introduction sections (**LOA 5**) as you suggested, and supplemented the principles, methods, procedures of the method we used to detect intracellular ROS in section 2.7.2, as listed in **LOA 8**, and the definitions of DEF and SER

in section 2.8.3, as listed in **LOA 9**. In addition, we have almost revised the descriptions of results including cytotoxicity, hypoxia sensitivity, cell cycle analysis, ROS generation and radiotherapy enhancement effect of the NCs, as **LOA 10**. Also, we have rewritten the discussion sections as you suggested, and detailly discussed the experimental results, additive effect between the two drugs, cell-type dependence phenomena (including the difference of hypoxia response), radiotherapy enhancement effect and the limitations of the present study, as listed in **LOA 11**. We want to say thank you for your careful review, patient correction and thoughtful comments. Thank you very much for the advices which help me revise my manuscript. Finally, we appreciate the main question you have raised in the e-mail that **“if we were to target 5FU as suggested by this paper then its role to attack metastases will be removed”**, which has pointed out the metastatic situation that we had lost sight of. Moreover, we address the question as follows:

Firstly, we need to clarify that 5-FU is loaded onto the nanocomposites (NCs) via physisorption (last sentence of paragraph 3, section 3.1, page 7), it is neither the target of the NCs nor targeting transported by NCs. We are sorry for the misunderstand.

Secondly, **with regard to this issue**, we should recognized that it's our thoughtless for losing sight of the metastatic situation, and thank you for raising that point. Nevertheless, the physical loading of chemotherapeutics doesn't alter the structure of the drugs so it does not influence its antitumor effect as well. Once being released, it is no different from normal drug, therefore, its role in attacking cancer cells doesn't remove. Finally, in the case of remote cancer cells, it is possible that the antitumor effect will be limited because the distance makes the concentration of released drugs too low to inhibit the remote cancer

cells. However, this problem can be resolved through rigorous quantitative studies of NCs releasing properties, knowledge of pharmacokinetics and a variety of in vivo experiments. Therefore, we can't say that the drug effect will be removed by carrier loading.

In addition, we have **listed some references** about other researchers' studies on drug delivery system loading 5-FU for cancer treatment [1, 2]. Moreover, we have also listed other novel delivery systems which deliver RNA interference [3] for gen-silencing cancer therapy, or insulin for diabetes treatment [4], and antigen for cancer immunotherapy [5], which all demonstrate that the load does not affect drug efficacy.

Moreover, the **answers to the questions and suggestions listed in the attached file** are moved to part 3.

That's all. Thank you again for your serious consideration and constructive comments. We appreciate your advices sincerely.

Reference

[1] T. Scharnweber, C. Santos, R. Franke, M. M. Almeida and M. E. V. Costa. Influence of Spray-dried Hydroxyapatite-5-Fluorouracil Granules on Cell Lines Derived from Tissues of Mesenchymal Origin, *Molecules* 2008, 13, 2729-2739; DOI: 10.3390/molecules13112729.

[2] J. L. Arias. Novel Strategies to Improve the Anticancer Action of 5-Fluorouracil by Using Drug Delivery Systems. *Molecules* 2008, 13, 2340-2369; DOI: 10.3390/molecules13102340.

[3] H. M. Liu, Y. F. Zhang, Y. D. Xie, Y. F. Cai, B. Y. Li, W. Li, L. Y. Zeng, Y. L. Li, R. T. Yu, hypoxia-responsive ionizable liposome delivery sirNa for glioma therapy. *International*

Journal of Nanomedicine, 2017, 12, 1065–1083. (<http://dx.doi.org/10.2147/IJN.S125286>)

[4] J. C. Yu, Y. Q. Zhang, Y. Q. Ye, R. DiSanto, W. J. Sun, D. Ranson, F. S. Ligler, J. B. Busec, and Z. Gu. Microneedle-array patches loaded with hypoxia-sensitive vesicles provide fast glucose-responsive insulin delivery. *PANS*, 2015, 112, 8260-8265. (www.pnas.org/lookup/suppl/doi:10.1073/pnas.1505405112/-/DCSupplemental)

[5] Y. Z. Min, K. C. Roche, S. M. Tian, M. J. Eblan, K. P. McKinnon, J. M. Caster, S. J. Chai, L. E. Herring, L. Z. Zhang, T. Zhang, J. M. DeSimone, J. E. Tepper, B. G. Vincent, J. S. Serody and A. Z. Wang. Antigen-capturing nanoparticles improve the abscopal effect and cancer immunotherapy. *Nature nanotechnology*, 2017(26), DOI:10.1038/NNANO.2017.113

Reviewer: 2

The Reviewer's comments:

In this paper, the authors reported a promising novel multifunctional platform which displayed a self-amplified effect that activated chemoradiotherapy co-enhancement. The research is interesting. And there are some suggestions to improve this article.

1) I think the words "panting efficiency (PE)" (Page 5, line 41) should be changed to "plating efficiency (PE)" and the words "the sensitization enhancement radio (SER)" (Page 13, line 8-9) should be changed to "the sensitization enhancement ratio (SER). Please check carefully.

2) What is the meaning of the word "SF2" in the manuscript? Maybe it should be changed to "SF" (survival fraction).

3) From the Figure 6a, we know the value of S of H1975 cells treated by MGO/FU is 28.7 and that of S of H1975 cells treated by MGO/FU-MI NCs is 27.8. But in Figure 6b, we observe the values represented by the graph are different which the value of S of H1975 cells treated by MGO/FU-MI NCs is larger than that of S of H1975 cells treated by MGO/FU. Maybe the authors should pay attention to this.

4) Please check the manuscript carefully.

The authors' responses:

Firstly, we intend to thank Reviewer: 2 for saying that "**The research is interesting**".

Moreover, thank you for your careful reading and patient correction of my previous manuscript. We appreciate your suggestions about spelling and data presentation, which

help us a lot in improvement of our manuscript. In addition, we have addressed all the points raised as summarized below.

- 1) We follow the recommendation to replace the **spelling** of “panting efficiency (PE)” with “planting efficiency (PE)” (line 9, section 2.8.1, page 5) and “the sensitization enhancement radio (SER)” with “the sensitization enhancement ratio (SER)” (line 5, paragraph 2, page 10), as listed in **LOA 15**. We are sorry for the spelling errors. Thank you for your correction and we have checked the manuscript carefully this time.
- 2) The word “**SF₂**” refers to the **SF under 2 Gy**, which is the common clinical dose fractionation. Thank you for point out the unclear expression, and we have emphasized the meaning in experimental section (last line of section 2.8.2) and added a special **footnote** below the table 1, as listed in **LOA 9&14**.
- 3) Thank you for raising the **inconformity of my data presentation in figure 5a&b**. It is our thoughtless for neglecting to explain the relationship of the two Figures. Specifically, Figure 5a exhibits one of the results of cell cycle analysis on H1975, while Figure 5b exhibits the statistical results of three independent experiments of cell cycle analysis on H1975. Therefore, the inconformity comes from the accidental errors between experiments. And we have emphasized the relationship of the two figures in captions to make it clearer as listed in **LOA 14**. Thank you for your careful consideration.
- 4) We have totally rewritten the previous manuscript and **checked it carefully** once again, thank you for your advices.

The above is all of our answers, thank you again for your patient correction and thoughtful consideration, we appreciate your comments very much.

Reviewer: 3

The reviewer's comments:

In this manuscript, the authors discussed the development of a multifunctional graphene oxide based nanocomposites to deliver metronidazole, 5-fluorouracil, and PePt magnetic nanoparticles for chemoradiotherapy. The integration of hypoxic drug carrier, radiosensitizer and chemotherapeutic drugs confers hypoxia-sensitivity and chemoradiotherapy co-enhancement functions to a single nano-platform. The detailed in vitro studies of the nanocomposites using A549 and H1954 cells exhibit significant cell proliferation suppression, intracellular ROS burst, and improved radiosensitivity, which could act as a potential multifunctional nano-platform for tumor treatment to improve the synergistic effect of chemoradiotherapy. However, some important parameters of this nano-platform that could affect the in vivo application and translation for nano-drug delivery systems are missing. There are some major concerns associated with the current version of submission which are listed below.

Comments:

1. The PePt MNPs and 5-FU were loaded on the MGO/FU-MI NCs via physisorption. One concern with the physical loading systems is the cargo loading stability and release kinetics. It is known that the cargoes may release during circulation, which leads to low drug delivery to the tumors, poor treatment efficiency and potentially toxicity to normal tissues. It is important for the authors to add the loading stability of the NCs and the release profile in both physiological and tumor environment condition.

2. For any drug delivery system, the loading efficiency and loading capacities are critical for effective tumor therapy. These data are missing for the MGO/MI, MGO/FU, and MGO/FU-MI nanocomposites. Additionally, it is useful for the audience to understand the advantages of this system by showing the data regarding the optimal cargo loading amount and ratio since these data may affect the cell viability and ROS production results.

3. It is confusing that the cancer cells showed reduced cell viability under normoxia condition in the presence of MGO/FU-MI NCs in Figure 4g. Typically, the hypoxia sensitivity of NCs should facilitate cell proliferation suppression under hypoxia condition.

4. In page 12 line 4: "The composites loading 5-FU showed an S phase arresting, the fraction of which was nearly one-times augment for H1975 cells cultured with MGO/FU NCs." Is it a typo that the "MGO/FU" should be "control"? As the composites loading 5-FU should also be MGO/FU NCs. And, "one-times" should be "one-time".

5. In figure 2f, what are the two peaks at 9 keV and 11.4 keV attributed to?

The authors' responses:

We would like to thank Reviewer: 3 firstly for the careful review and thoughtful consideration on our previous manuscript. The suggestions about **supplementing the loading characters** of the nanocomposites (NCs) are valuable and constructive. Just as you said, an unstable loading system would cause normal tissue toxicity and low drug efficiency. Moreover, the loading efficiency and capacities is also of great important of a loading system for it will not only influence the degree of in vivo toxicity of the NCs but also relate to the advantages of the carrier. They are the very assignments we plan to do for the quantitative examination and in vivo application in the further study. However, as the

present study was preferred to qualitative examination, some raw materials in the present synthesis are somewhat superfluous added and the consumption of which is needed to be optimized further for the better quantitative application in vivo. Therefore, as we have detected, with the current raw materials input ratio, the loading efficiency of MI and 5-FU is not satisfactory. Here is our point by point response:

- 1) With regard to the concern of **cargo loading stability and release kinetics**, we have supplemented the results and descriptions of release profiles of MI and 5-FU in Figure 3d&e and section 3.1 (paragraph 4, page 6-7), as listed in **LOA 7**. The deionized water and PBS solutions containing 500ug/ml NCs were used and incubated for 24 hours to simulate the physiological and tumor condition respectively. As shown in Figure 3d&e, with 24hours' incubation in H₂O, the concentrations of MI varies from 0.4 ~ 2.1ug/ml, while it is 1.9~4.2ug/ml for 5-FU. The results is similar to those in PBS solutions. The release is little for 500ug/ml NCs, and it would be less for the experiments concentration (20ug/ml), which means a negligible cytotoxicity is caused by drug releasing with 24 hours' incubation.
- 2) In terms of **loading efficiency and loading capacities**, we have examined the loading efficiency of MI and 5-FU using UV-Vis spectrophotometer, as listed in **LOA 7**. According to the data (line 2, paragraph 4, page 6), the loading efficiency of the two drugs is low, with MI of 25.03% and 5-FU of 27.76%. However, as we have mentioned above, there is a surplus of the raw materials and it still needed optimization for the further quantitative application in vivo. Therefore, we cut the consumption of MI and 5-FU in half and examine the loading efficiency again to verify whether it is redundant.

This time, the loading efficiency of MI increases to 57.24%, indicating that the consumption of MI is indeed redundant. However, the loading efficiency of 5-FU is reduced to 17.44%, which means that the loading capacities of 5-FU is low, and that GO needs to be further modified to improve the loading capacity of 5-FU.

Moreover, for **optimal cargo loading amount and ratio**, multiple tests with different raw materials ratio and the consequent biological effect is needed, which are too time-consuming to confirm. We are very sorry that we can't provide such data within a short time. Therefore, the optimal cargo loading amount and ratio would be implemented in our further study. Thank you for your constructive suggestions, which have greatly helped us revise our study. Moreover, we will try our best to complete it in the following study.

- 3) Thank you for pointing out the confusing results of **cell viability reduced under hypoxia condition in Figure 4g**, which has prompted us to think and discuss in depth and thus improved our manuscript in indirectly. Moreover, we have supplemented the possible reasons and mechanisms in section 3.2 (paragraph 3, page 8) and discussion section (line 3-12, paragraph 3, page 11), as listed in **LOA 10&11**. Also, we have pasted it as follow: "The hypoxia sensitivity tests is vary an example of that (**cell-type dependence**). As it exhibits (Figure 4g), the cell viability of H1975 cells treated with MGO/FU-MI under hypoxia condition do not decrease but increase compared to normoxia condition, while the result of A549 cells is contrary. The difference suggests that different cells would activate diverse regulatory mechanisms when exposed to hypoxia environment, which would influence the cytotoxicity of NCs accordingly. In addition, comparing to the result of H1975 cells with MGO-MI treatment, the adding of 5-FU should account for the difference likewise. In another words, the interactions between 5-FU and MI would stir up some intracellular signaling pathways that could

influence the hypoxia-response of MI and thus lead to the failure or reduction of hypoxia sensitivity of the NCs upon H1975 cells. However, all speculations are speculative and require further pharmacokinetic investigations.” **We hope that we have made the problem understandable and thank you for your comments again.**

- 4) **We have rewritten the sentence as** “Compared with the control group, MGO/FU NCs shows an S phase arresting effect, the fraction of which is almost one-time enhanced on H1975 cells.” (line 1, page 9), as listed in **LOA 15**. Thank you for your correction.

- 5) Thank you for raising the point of the omission of notations in **Figure 2f**, which has reminded us of that more attention should be paid to details. We have supplemented it, as listed in **LOA 14**. As we have known, each element has a cluster (not one) of characteristic energies due to different energy level transitions, so there will be a lot of characteristic energy peaks belonged to one element in the EDS spectrum. And the peaks in the spectrum at 1.7 and 11.4 keV belong to Pt while the peak at 9 keV belongs to Cu. The results have supplemented in Figure 2f. Thank you for your comments again.

That’s all of my answers. We appreciate your constructive comments which have greatly help us revise our manuscript. Moreover, the comments also let us recognize the limitations of our present study, and we will try our best to complete it in the further study.

Thank you for your careful review and constructive comments again.

Part 3: Answers to Questions Listed in Attachment form Reviewer 1.

General;

1. The 5FU is meant to go around whole body and affect the DNAs of cancer cells including metastasis. If it is i.e. the 5-FU targeted, then it will miss its effects on remote cancer cells.

Thank you for your valuable comments, and we have answered this question in Part 2, so we will answer it briefly here.

Firstly, 5-FU is loaded onto the nanocomposites (NCs) via physisorption (last sentence of paragraph 3, section 3.1, page 7), it is neither the target of the NCs nor targeting transported by NCs.

Secondly, the physical loading of chemotherapeutics doesn't alter the structure of the drugs so it does not influence its antitumor effect as well.

Finally, in the case of remote cancer cells, it is possible that the antitumor effect will be limited because the distance makes the concentration of released drugs too low to inhibit the remote cancer cells. However, this problem can be resolved through rigorous quantitative studies of NCs releasing properties, knowledge of pharmacokinetics and a variety of in vivo experiments. Therefore, we can't say that the drug effect will be removed by carrier loading.

In addition, the references related to drug delivery system loading 5-FU for cancer treatment [1, 2].and other novel delivery systems which delivered RNA interference [3] for gen-silencing cancer therapy, or insulin for diabetes

treatment [4], and Antigen for cancer immunotherapy [5], are listed again, which all demonstrate that the load does not affect drug efficacy. Thank you again for raising that problem, making us think more deeply.

2. Any results to be shown in the abstract??

Thank you for the suggestion and we appreciate your comments very much.

Through showing the key points of results in the abstract, highlighting the advantages of our materials and making it clearer for audiences to comprehend it. Therefore, we have partially rewritten the abstract section (line 9-14 of abstract, page 1) according to the comments, as listed in **LOA 4**.

3. There is no word as “Leaded” please replace it through out the manuscript into “Lead” or “Led”

We are sorry for the error and we have replaced the word “leaded” into “lead” as suggested, Thank you for your correction.

4. The method for measuring the levels of generated ROS as briefly mentioned in section 2.6.2 is not clear and not explained to the reader to understand how are the ROSs detected and counted??

Your comment on the above issue is greatly appreciated. Through supplementing the principles, methods and procedures of it in section 2.7.2 (page 5) as suggested and listed in **LOA 8**, we have found the manuscript is more complete and clearer for the audiences to comprehend the study. Thank you for your advice and we have pasted that part as follows: “The effects of NCs on intracellular ROS generation were examined by ROS assay kit (Beyotime, China), which uses dichloro-dihydro-fluorescein diacetate

(DCFH-DA) fluorescence probe to detect intracellular ROS. Specifically, DCFH-DA itself has no fluorescence and can pass through the cell membrane freely. After entering the cell, it can be hydrolyzed to DCFH by intracellular esterase, which can't penetrate the cell membrane, making it easy for becoming the prob to be loaded into the cell. Moreover, ROS in the cells can oxidize non-fluorescent DCFH and produce fluorescence DCF. Therefore, the level of intracellular ROS could be knew through measuring the fluorescence intensity of DCF in cells. In the experiments, 3×10^5 cells were seeded in 6-well plates and cultured overnight. Then cells were treated with MGO-MI, MGO/FU, MGO/FU-MI NCs solutions with final concentrations of $20 \mu\text{g/ml}$ for each well, and incubated for 24 hours. Next, cells were washed, trypsinized, centrifuged and resuspended in 10mM DCFH-DA serum-free medium solution and incubated for 30 mins at 37°C without light. After that the cells were washed twice with serum-free medium and collected for fluorescence analysis using a flow cytometer (Cytomics TM FC 500, Beckman Coulter, Inc., USA) and 10,000 events were counted for each sample. The data were analysed using FlowJo and GraphPad Prism software. The results were expressed as means \pm SD"

Corrections;

ABSTRACT:

1. Line 4; "drug delivery system with 246nm" what does this mean? Is this diameter? If so of what?

Thank you for your comment that reminds us to express clearer. We have changed the expression to "with an average size of 243nm" (line 4, abstract section) as suggested and listed in **LOA 4** to make it clarity. It is the average size of the nanocomposites (NCs) and the numerical change is consistent with the result of DLS in Figure 2c, which is our carelessness to have made the numerical error.

2. Line 7 “Chemo-therapeutics “ remove ‘s” at the end

We have changed “chemotherapeutics drugs” to “chemotherapeutic drugs” (line 7, abstract section) as suggested and listed in **LOA 4**, and thank you for your correction.

INTRODUCTION

1. Page 2- line 7- “attention that fabricating” replace “that” by “to”

Thank you for your grammatical correction, and we have rewritten this sentence as “the idea of developing a hypoxic drug-delivery system loading anticancer agent and radiosensitizers simultaneously to improve the efficiency of tumour chemoradiotherapy is popular with researchers” (line 5-7, page 2).

2. Second paragraph: line 6; “to delivery insulin for ..” change “delivery’ to “Deliver”

We have changed “delivery” to “deliver” (line 6, paragraph 2, page 2) as suggested and listed in **LOA 5**. Thank you for your correction.

3. Same line as above: “where”? does not make sense. Change the whole sentence o may be replace ‘where” with “were”?

Thank you for the comment and we have rewritten the sentence as “Ji-Chang Yu et al. developed a hypoxia-sensitive vesicle to deliver insulin for diabetes therapy, using NI as hypoxic ingredient likewise” (line 5-6, paragraph 2, page 2), which means “Ji-Chang Yu et al. utilized NI as raw material to develop the hypoxia-sensitive vesicle”.

4. Third paragraph; second sentence: Change “In especial” to “Especially”

We have changed “In especial” to “Especially” (line 2, paragraph 3, page 2) as suggested and listed in **LOA 5**. Thank you for your correction.

5. Figures 2-a and 2-b are not clear and the scale is unreadable

Thank you for bringing up the problem which makes us pay more attention to the readability of the Figures. However, scale bars in Figure 2a&b are unable to change, because they are automatically generated when taking pictures with TEM. Therefore, we have stressed them in the captions to make it clearer for audiences, as listed in **LOA 14**.

METHOS:

1. Section-2.2.1 second sentence is not clear needs re-writing

We appreciate the comment and have rewritten the sentence as “Briefly, OA (1.5ml), OL (1.5ml) and Fe(acac)₃ (0.386mmol) were dissolved in anhydrous ethanol (100ml) and stirred for 30 mins. Then, H₂PtCl₆ • 6H₂O ethanol solution (20 ml, 19.3mmol/L, 0.386mmol) was transferred into the mixture mentioned above and stirred for another 30 mins” (line 2, section 2.2.1), as listed in **LOA 6**.

2. Section 2.2.2 Replace “500” by a word such as “The” etc...

We have changed the expression to “The PEG (500mg, 0.25mmol) was dissolved...” (first line, section 2.2.2).

3. Section 2.3 “Dynamic light scatter” change to “Dynamic Light Scattering device”

We have changed “dynamic light scatter” to “dynamic light scattering device” (line 3, section 2.3) as suggested. Thank you for your correction.

4. Section 2.6.2 Explain what is meant by “DCF”

As we have answered above, the principles of the dichloro-dihydro-fluorescein diacetate (DCFH-DA) fluorescence probe used in the study to detect intracellular ROS (line 3-8, section 2.7.2, page 5) have been supplemented. Moreover, we would like to explain again that DCF is the fluorescence production during the detecting process, the intensity of which is just proportional to the content of intracellular ROS. Therefore, we detected the fluorescence intensity of DCF to represent the level of ROS.

RESULTS:

1. Section 3.1 second sentence: remove “At the meantime” and replace it with ‘The”

We have replaced “At the meantime” with “the” (line 3, section 3.1, page 7) as suggested.

2. Same sentence; Following FePt MNPs replace “was” with “were”

We have changed “was” to “were” (line 3, section 3.1, page7) as suggested, and thank you for your correction.

3. Section 3.1 line 9 : change “exhibited” to “exhibits”

We have changed “exhibited” to “exhibits” (line2, paragraph 2, section 3.1, page 7) as suggested.

4. Similarly few other words need to be changed for ending into “ed’ into “s”

Your comments on checking the verb tense throughout the manuscript is good and appreciated greatly. The problem of tense had puzzled us deeply,

and we have changed it to simple present tense when describing the result and conclusion. The past indefinite tense is utilized to describe the experiment procedure as suggested and listed in **LOA 16**. Thank you very much for your correction.

5. Section 3.2 line 4: change 'cargos loaded' to "loaded cargos"

We have changed sentence to "and some of them have degraded and released the cargos" (line 3, section 3.2 page 8).

6. Page 10: Font sizes in some figure 4 sections are not readable.

Thank you for pointing out the problem of the readability of the Figures again, we appreciate the suggestions and we have tried to enlarge the font size in Figure 4 to make it clearer for audiences, as listed in **LOA 14**.

7. Also in page 10. The third line from bottom of the page; "replace "responsive" with "response"

We have replaced "responsive" with "sensitivity" (line 2, paragraph 3, section 3.2) and thank you for your correction.

8. There is no explanation for the results of Hypoxia response differences between the 2 type of cells??

Thank you for pointing out the difference results of hypoxia response between the two type of cells, the comment is good, which promotes us to think and discuss carefully to address the question. Therefore, We have totally written the part of hypoxia sensitivity tests, and carefully explained the reasons for different results (paragraph 3, section 3.2, page 8) as suggested and listed in

LOA 10. In summary, through comparing the contrary results of H1975 and A549 cells treated by MGO/FU-MI NCs, we believe that hypoxia condition would stir up diverse cellular regulatory mechanism which includes the uptake and metabolic mechanisms of cells, influencing the outcome of cytotoxicity accordingly. Moreover, through comparing the results of H1975 cells treated by MGO-MI and MGO/FU-MI NCs, the additive effect between MI and FU also should account for the phenomenon. It implies that the interaction between the two drugs would stimulate some cellular signal pathway, which could lead to the failure or reduction of the hypoxia sensitivity of MI. We have also supplemented that part in discussion section (line 3-13, paragraph 3, page 11), as listed in **LOA 11**, in expectation of making it comprehensible for audiences.

9. The second sentence on page 11 is not clear. Please re-write it. Something as "...Hypoxia-response was cell difference" does not make sense. Do you mean that Hypoxia depends on cell type?

Thank you for taking out the confusion and we have totally rewritten this part to make it clearer (paragraph 3, section 3.2, page 8), as listed in **LOA 10**. In addition, we must explain that the statement of "Hypoxia-response was cell difference" didn't mean that Hypoxia depends on cell type. What we wanted to tell was that the hypoxia condition would stimulate different coping mechanisms in different cell types, such as regulation of uptake and metabolic mechanisms to adapt to the new environment. And the different

coping mechanisms would make a great difference on drug cytotoxicity.

Therefore, we had said that “Hypoxia-response was cell difference”. Now we have deleted the sentence and rewritten the paragraph. Thank you very much for your good comments on that part, which help us revise our manuscript greatly.

10. Section 3.3: First sentence; add “for” before 24h

We have added “for” before 24h (line 1, section 3.3, page 8) as suggested.

11. Same sentence; add “concentration of” before 20

We have added “concentration of” before 20 (line 1, section 3.3, page 8) as suggested

12. Also this sentence claims that ROS is measured. But How? At least refer to the method. How reliable it is etc..

Thank you for mentioned the reliable of the method we used, therefore, we have looked for references which mentioned and utilized the method[6, 7, 8], and detailed studied and discussed the mothed totally[9, 10], as listed in **LOA**

13. Through this process, we have get more knowledge on the method, and knew that it also has limitations in detecting ROS [9, 10]. We benefit a lot from this comments, which make us realize the limitations and the rang of application and the degree of accuracy of a experimental method. It also reminds us of the significant of deep investigation before a study. Thank you very much for your comments and we have supplemented the principles,

methods, and procedures of the method utilized to measure ROS in the study in section 2.7.2 (page 5), as listed in **LOA 8**.

13. The sentence in the middle of the second paragraph page 12: “As depicted in figure 5d,...” There is no explanation for this observation????

Thank you for your feedback, which promotes us to totally rewrite the sentences and descriptions of this part (line 4-13, paragraph 2, page 9) to make the results of ROS detection clearer and more comprehensible for the audiences, as listed in **LOA 10**.

14. The following sentence mentions “burst of ROS” How? And why?

Thank you for mentioning the question again, the comments promote us to complete this part thoroughly, and our manuscript has improved a lot through the rewriting, as listed in **LOA 8&10**. In summary, the principles, methods, and procedures of the method utilized in our study to detect intracellular ROS have been supplemented in section 2.7.2 (page 5), in which we have known that the fluorescence intensity of DCF is directly proportional to the ROS content. Moreover, the descriptions of the ROS detection results (line 4-13, paragraph 2, page 9) have been totally rewritten, in which we have known that the fluorescence intensity is obviously augmented after the NCs treatment, suggesting the ROS augment caused by NCs treatment.

15. This section 3.3 is very shallow it needs expansion and further explanation & discussion of the results

All of the comments related to this points are good and greatly appreciated, we benefit from it a lot. Moreover, our manuscript has improved a lot according to these comments. Thank you very much for your careful review and thoughtful comments, and we have summarized the actions we have made on this part as follows: Firstly, we have supplemented the principles, methods, and procedures of the method in section 2.7.2 (page 5) to make it clearer (**LOA 8**). Then, totally rewriting of the descriptions of the results (line 4-13, paragraph 2, page 9) is made to make the results comprehensible (**LOA 10**). Finally, the results are discussed slightly in the discussion section(paragraph 1-2, page 11) to stress the meaning of the results (**LOA 11**).

16. Section 3.4 is very poorly written. It is not the matter of language only but structure and material. Many sentences don't make sense at all. For instance the second one which starts with "As expected.." as expected based on what??. I tried to help organising the section but I realised that I will end up re-writing it which is not my responsibility.

Thank you for your careful review and patient correction, your comment is good and appreciated, and we have totally rewritten the 3.4 section (page 9-10) to make it better organized (**LOA 10**). Specifically, the section 3.4 is divided into three parts. The first part exhibits the effect of radiation enhancement of the NCs under 2Gy, the common clinical dose fractionation, which displays the great application prospects of the NCs (paragraph 1,

section 3.4, page 9) primary. Moreover, the radiotherapy enhancement effect of the NCs is proved in the second part, in which dose enhancement factor (DEF) is utilized to evaluate the dose efficiency enhancement in dosiology (paragraph 2, section 3.4, page 9-10). Finally, the biological mechanisms of the radiotherapy enhancement effect are explored in the third part. In which cell radiation sensitization ratio (SER) is utilized to evaluate the biological sensitivity enhancement effect of the NCs in radiobiology (paragraph 2, page 10). Therefore, with the description of the three aspects, we have displayed that the NCs will not only improve the radiation efficiency but also improve the cell radiosensitivity, which would be a good radiation sensitizers.

17. Figure 6-b should be marked 2Gy to compare it with a

Thank you for your suggestion which improves the readability of our figures. Therefore, we have changed the title of the Figure 6b to 2Gy for comparison with Figure 6a as suggested and listed in **LOA 14**.

18. Figure 6-d; how is this curve fitting done? Can it be shown what degree of polynomial is used? As it does not look right

Your comment of this part is good and appreciated as well, which improves our manuscript again. Therefore, we have totally rewritten section 2.8 in experimental section (page 5-6) to make the experiment tools more comprehensible for audiences (**LOA 9**). Specifically, we have supplemented the equations and definitions of SF_2 , DEF, SER, the “multi target-single

hitting” model, and the meanings of D_0 and N in radiobiology. Moreover, as for this question, our answers are as follows:

The survival curve was fitted with the “multi target-single hitting” model, which equation is as follow. In addition, the survival fraction is displayed on logarithmic coordinates. The spots in figure 6d are the experimental data obtained from clonogenic survival assays while the curves are the functions obtained through fitting the data using GraphPad Prism software.

$$SF = 1 - (1 - \exp(-D/D_0))^N$$

As you have mentioned, the spots deviate too far from the curve, making it looks incorrect. For this problem, we have to explain the fact that the cells in the experiments with 6, 8 and 10Gy were died for the most part and the SF were vary greatly as one or two clones count errors, so the data of small dose was accurate and significative. additionally, the logarithmic coordinates have magnified the difference hundreds and thousands of times between the spots and the curve. Actually, the difference were both smaller than 0.05, 0.01 or even.

19. DEF values listed in figure 6 are not defined and not explained how are they obtained?

Thank you for your comments, as we have mentioned above, the definition and equation of DEF are supplemented in 2.8.3 section (page 6), which has improved our manuscript a lot (LOA 9). Moreover, we have pasted it below, in which “ $D_c(SF_{90})$ ” represented the dose producing 90% SF for the control

group, while the $D_d(SF_{90})$ was the dose producing 90% SF for the drug group". The correspondent values were obtain through the survival curve.

$$DEF = \frac{D_c(SF_{90})}{D_d(SF_{90})}$$

20. Two terms DEF and SER are used through out the paper it needs some explanation as to what is the difference between the two terms if any

Thank you for your comments, and the definitions and equations of the two terms have supplemented in section 2.8.3 (page 6) as suggested and listed in LOA 9. Moreover, we have pasted it below for comparison purposes.

$$DEF = \frac{D_c(SF_{90})}{D_d(SF_{90})}; \quad SER = \frac{D_{0c}}{D_{0d}}$$

"Where $D_c(SF_{90})$ represented the dose producing 90% SF for the control group, while the $D_d(SF_{90})$ was the dose producing 90% SF for the drug group. Similarly, D_{0c} and D_{0d} were the parameters D_0 obtained in the control group and the drug group respectively."

From the equation above, we can know that, DEF as the dose enhancement factor emphasizes the enhancement of dose efficiency. In addition, SER as cell sensitization enhancement ratio, emphasizes the enhancement of cell radiosensitivity. The former comes from dosiology while the letters from radiobiology. The two terms represent the radiotherapy enhancement of the NCs from different perspectives.

21. Section 4 "Discussion" is poorly written, ambiguous, and about the results but about the idea of the project. Discussion should be about the results and what they mean and what are their limitations etc..

Your comment and suggestion on discussion section are constructive and appreciated. We have almost rewritten this section (page 10-11) according to your suggestion, which has included the discussion of cytotoxicity (paragraph 2), results of antitumor analyses (paragraph 2), additive effect between MI and 5-FU (paragraph 3), cell-type dependence phenomenon (paragraph 4), which includes discussions on results of hypoxia sensitivity (line 3-13, paragraph 3, page 11), radiotherapy enhancement effect (paragraph 5) and the limitations (paragraph 6) of the present study (**LOA 11**). The manuscript has revised a lot as you suggested. Thank you for your careful review and construction advices again.

22. The conclusion "section 5" also will benefit greatly from re-writing. Example; The sentence before the last: "future investigation was needed" this sentence doesn't make any sense. If it is future how is it "was"??

Thank you for your suggestion on section 5, and we have partially rewritten this part (page 12) as you suggested to make it clearer and better organized, covering the main points of the study. Moreover, we have changed the tenses as you suggested to make it more formal and correct (**LOA 12**).

Finally, thank you for your patience correction, careful review and profound thinking on our study. We have benefited a lot from your comments and suggestions. Also, our manuscript has greatly improved according to your comments, which is clearer, better organized, more complete and comprehensible for audience. In addition, we have obtained more knowledge

and indoctrinations which is of profound significance in our further study.

Thank you very much again.

Reference

- [1] T. Scharnweber, C. Santos, R. Franke, M. M. Almeida and M. E. V. Costa. Influence of Spray-dried Hydroxyapatite-5-Fluorouracil Granules on Cell Lines Derived from Tissues of Mesenchymal Origin, *Molecules* 2008, 13, 2729-2739; DOI: 10.3390/molecules13112729.
- [2] J. L. Arias. Novel Strategies to Improve the Anticancer Action of 5-Fluorouracil by Using Drug Delivery Systems. *Molecules* 2008, 13, 2340-2369; DOI: 10.3390/molecules13102340.
- [3] H. M. Liu, Y. F. Zhang, Y. D. Xie, Y. F. Cai, B. Y. Li, W. Li, L. Y. Zeng, Y. L. Li, R. T. Yu, hypoxia-responsive ionizable liposome delivery sirNa for glioma therapy. *International Journal of Nanomedicine*, 2017, 12, 1065–1083. (<http://dx.doi.org/10.2147/IJN.S125286>)
- [4] J. C. Yu, Y. Q. Zhang, Y. Q. Ye, R. DiSanto, W. J. Sun, D. Ranson, F. S. Ligler, J. B. Busec, and Z. Gu. Microneedle-array patches loaded with hypoxia-sensitive vesicles provide fast glucose-responsive insulin delivery. *PANS*, 2015, 112, 8260-8265. (www.pnas.org/lookup/suppl/doi:10.1073/pnas.1505405112/-/DCSupplemental)
- [5] Y. Z. Min, K. C. Roche, S. M. Tian, M. J. Eblan, K. P. McKinnon, J. M. Caster, S. J. Chai, L. E. Herring, L. Z. Zhang, T. Zhang, J. M. DeSimone, J. E. Tepper, B. G. Vincent, J. S. Serody and A. Z. Wang. Antigen-capturing nanoparticles improve the abscopal effect and cancer immunotherapy. *Nature nanotechnology*, 2017(26), DOI:10.1038/NNANO.2017.113
- [6] Y. M. Sun, H. T. Miao, S. J. Ma, L. Zhang, C. C. You, F. Tang, C. Yang, X. L. Tian, F.

Wang, Y. Luo, X. J. Lin, H. Wang, C. Y. Li, Z.J. Li, H. G. Yu, X. F. Liu, Y. Xiao, Y. Gong, J. H. Zhang, H. Quan, C. H. Xie. FePt-Cys nanoparticles induce ROS-dependent cell toxicity, and enhance chemo-radiation sensitivity of NSCLC cells in vivo and in vitro. *Cancer Letters*, 2018, 418, 27-40. (<https://doi.org/10.1016/j.canlet.2018.01.024>)

[7]C. P. LeBel, H. Ischiropoulos and S. C. Bondy. Evaluation of the Probe 2',7'-Dichlorofluorescein as an Indicator of Reactive Oxygen Species Formation and Oxidative Stress *Chem. Res. Toxicol.*, 1992, 5, 227-231.

[8]Y. Z. Chang, L. Z. He, Z. B. Li, L. L. Zeng, Z. H. Song, P. H. Li, L. Chan, Y. Y. You, X. F. Yu, P. K. Chu and T. F. Chen. Designing Core-Shell Gold and Selenium Nanocomposites for Cancer Radiochemotherapy. *ACS Nano*, 2017, 11, 4848-4858. (DOI: 10.1021/acsnano.7b01346)

[9] B. Kalyanaraman, V. D. Usmar, K. J.A. Davies, P. A. Dennerly, H. J. Forman, M. B. Grisham, G. E. Mann, K. Moore, L. J. Roberts II, and H. Ischiropoulos. Measuring reactive oxygen and nitrogen species with fluorescent probes: challenges and limitations. *Free Radic Biol Med.* 2012; 52(1): 1–6. doi:10.1016/j.freeradbiomed.2011.09.030.

[10] A. Aranda, L. Sequedo, L. Tolosa, G. Quintas, E. Burello, J.V. Castell, L. Gombau. Dichloro-dihydro-fluorescein diacetate (DCFH-DA) assay: A quantitative method for oxidative stress assessment of nanoparticle-treated cells. *Toxicology in Vitro* 27 (2013) 954–963. (<http://dx.doi.org/10.1016/j.tiv.2013.01.016>)

Appendix C

Dear Editor and Reviewers:

Thank you for the kind letter of decision and reviewers' comments on our manuscript, RSOS-181790 ("Development of a hypoxic nanocomposite containing high-Z element as 5-Fluorouracil carrier activated self-amplified chemoradiotherapy co-enhancement") on April 8th, 2019 again. The reviewers are all very insightful and able to raise the constructive comments, pointing out the limitations of our present research. We indeed appreciate the comments which make us more thoughtful and careful. Now we have revised the manuscript again, hoping to have addressed the questions clearly and get the approval of every reviewer. The Following is our description on revision, which is divided into two parts. Among them.

Part 1: List of Actions, which is a summary statement that covering all changes we have made in the revision.

Part 2: Authors' Response to Reviewers, which is our response to the comments from reviewers.

Part 1: List of Actions

LOA 1: We have changed the concentrations of 5-FU to 2mg/ml (page 3, section 2.2.4, line 2), which is a clerical error before.

LOA 2: We have standardized units for "ug/ml" to "µg/ml" and "ul" to "µl" throughout the manuscript.

LOA 3: We have supplemented the loading capacity calculation (page 4, section 2.4, line 9) and changed the releasing conditions of the NCs (page 4, section 2.4, line 11-13), as suggested by **Reviewer 3**. Therefore, the figures of releasing curves of MI and 5-FU in Figure 3, and the corresponding captions were changed as well. Moreover, we have almost rewritten the description of the results of loading and releasing characteristic (page 7-8, last paragraph)

LOA 4: We have rewritten some of the descriptions in section 3.2-3.4 to make the results more comprehensible for the readers (page 8-10).

LOA 5: We have almost rewritten the discussion section (page 10-13), organizing the structure to make it more comprehensible, as suggested by **Reviewer 3**. To be specific, we have divided the discussion into seven parts, which are research background (page 10-11, section 4, first paragraph), antitumor effect of the NCs (page 11, paragraph 2), the additive effect between 5-FU and MI (page 11, paragraph 3), the cytotype dependence phenomenon (page 11-12, last paragraph), the discussion of hypoxia sensitivity results (page 12, paragraph 2), the radiotherapy improvement mechanism of the NCs (page 12-13, paragraph 3) and the limitations of the present study (page 13, paragraph 2).

LOA 6: We have supplemented some references on chemoresistance in hypoxia cells (page 14-18, [47], [62-64]), as suggested by **Reviewer 3**, and reuploaded the support data (page 14, section 6).

Part 2: Authors' Responses to Reviewers:

Reviewer: 1

The reviewer's comments:

All aspects of this manuscript are cleared now. Except one point: the 5FU when used in cancer patients it is not only to affect the cancer volume but also if any metastasises that might exist anywhere within the body. Hence, targeting 5FU to the tumour volume will result into limitation of its applications. This point needs to be made clear by the authors for the readers. They attempted in their revision of the text but it is still not clear enough. I think the authors didn't understand the request and I hope now it is clear. So please address this issue before the publication of the paper

The authors' responses:

We want to say thank you first for your approval of our first revision, we owe it to your patient correction and thoughtful guidance during the first review. We appreciate your satisfaction with our manuscript. As for the situation of metastatic tumor, we must admit, as you have pointed out, that targeting 5-FU into the tumor volume does limit its antitumor effect on metastatic tumors that may exist anywhere within the body. Therefore, we have supplemented it as limitations of the present study in the discussion section (page 13, paragraph 2, line 5-18) (LOA 5) to make it clearer for the readers, and we have pasted it below. Thank you for your patient guidance.

"Firstly and constitutionally, the present drug delivery system should be further modified by biological target to expand the scope of its antitumor action. In conventional chemotherapy, chemotherapeutic drugs are to go around the whole body via intravenous injection to attack cancer cells, including metastatic tumors. The drugs themselves are not targeted, and their enrichment in the tumor sites is mainly due to the ubiquitous enhanced permeability and

retention (EPR) effect in solid tumors [68, 69]. However, the EPR effect always applies to macromolecular substance. Therefore, when it comes to small molecule chemotherapeutic drugs like 5-FU, they are not targeted at all, and they will attack all cells in the body, including cancer cells, metastatic tumors and normal tissues. As a result, the problem of systemic toxicity arises [7]. As a consequence, the hypoxic drug delivery system was designed and synthesized in the study to solve the problem of large dosage and low efficiency of clinical medication. However, every coin has two sides, the present designed is imperfect, which will limit its antitumor effect on metastatic tumors that may exist anywhere within the body. Therefore, further modification is required to expand its action scope of antitumor effect."

Lastly, we would like to thank you for all your work during these two reviews. Your insightful comments and guidance benefit a lot not only to our revision but also to our future study. We appreciate your advices very much and we will be more careful and thoughtful in the further study. Thank you very much.

Reviewer: 2

The Reviewer's comments:

The authors have further improved the manuscript. I think it can be accepted for publication now.

The authors' responses:

Thank you for your approval. It is very encouraging for us. We appreciate your suggestions and comments during these two reviews, and we will be more careful and diligent in the future study. Thank you very much.

Reviewer: 3

The reviewer's comments:

The authors added more details to the resubmitted version according to the comments. However, this version still have some issues that the authors need to address.

1) I didn' t clarify the physiological and tumor environment conditions, which are not water and PBS. I would like the author to compare the stability and release of the NPs at different pH, normally, 7 and 5-6. (doi: 10.1186/1475-2867-13-89). In addition, it will be more comprehensive to use cumulative release curve in Figure 6.

2) The authors added the loading efficiency of the NPs in the present synthetic method. However, I would prefer to know the loading capacity, which could represent the maximal loading dose of cargos on NPs, while the loading efficiency depends on the feeding amount.

Loading capacity (%) = (the weight of loaded drug/ the weight of NPs)

3) The author gave an explanation about the cell-type dependence phenomena, please add some reference support. However, the writing of the corresponding paragraph is quite confusing. For example, what is the meaning of the sentence "The hypoxia sensitivity tests is vary an example of that" ? Not only this paragraph, all the discussion section needs to be carefully revised. There are even many typos in the rebuttal letter.

4) 5) no further comments.

The authors' responses:

First of all, thank you for your constructive advice in these two reviews. We appreciate your suggestions about supplementing the loading and releasing characteristics of the nanocomposites (NCs) very much. They are indeed the most significant parameters needed for further drug delivery system evaluation and *in vivo* applications. Now, we have calculated the loading capacity of the NCs and supplemented it in manuscript (page 4, section 2.4, line 9), and changed the releasing conditions of the NCs (page 4, section 2.4, line 11-13) (**LOA 3**). Thank you for your suggestions which improve our study a lot. And we will be more thoughtful and careful in the further study. We address the questions as follows:

1. Thank you for pointing out the exact conditions of physiological and tumor environment.

Through investigation and survey, we have know that the acidity is also a major feature of microenvironment in tumor tissues, which is associated with a variety of physiological activities of tumors, such as proliferation, angiogenesis, immunosuppression, invasion and chemotherapy resistance. Knowing the releasing prolife of the NCs under acidic condition is of great significance for us to grasp the release characteristics of the NCs in tumor environment. Therefore, we have changed the releasing conditions of the NCs (page 4, section 2.4, line 11-13), and almost rewritten the descriptions of it (page 8, first paragraph, line 1-9) (LOA 3), as pasted below:

“Moreover, the releasing curves of MI are illustrated in Figure 3d. As shown, after 24 hours’ incubation, the MI is hardly released in neutral PBS. And even in acidic PBS, its release is slight, which is 0.3 μ g/ml. The result suggests the stability of the NCs even the acidic environment. In comparison, after 24 hours’ incubation, the release of 5-FU is relatively high, which is 2.8 μ g/ml in both neutral and sub-acid environments (Figure 3e). The difference in MI and 5-FU releasing characteristics may connect with the different loading methods of the two drugs. The former is covalently clinked to GO, while the latter is loaded onto GO via physisorption with the support of PEG, so it is relatively easier than MI release.”

As it can be seen that, due to the different loading methods of MI and 5-FU, the two drugs perform different releasing curves. In addition, the release amount is slight even in the acidic environment, suggesting the stability of the NCs.

In addition, we have recomposed the discussion of radiotherapy improvement mechanism of the NCs in combination with the release characteristics (page 12, paragraph 2, line 2-22) (LOA 5) to make it more comprehensible, and we have pasted some of it below. Thank you very much for the constructive suggestions, which improve our manuscript a lot. And we will be more thoughtful and careful in the future study.

“Therefore, in the design of the NCs, FePt MNPs will act as high-Z type radiosensitizers [6, 45], which enhance the radiation energy deposition within tumor through its high radiation rays absorption coefficient [13, 37] during the physical stage. Moreover, MI can assist the action of radical anions through simulating the action of oxygen [9, 65] in the chemical stage. Whereas the physical and chemical stage last short, the magnitude of the enhancement effect is depended on the amount of NCs in cells. As it has been proved by the releasing curve of the NCs that it is very stable in solutions. Therefore, with the uptake of cells, the NCs will perform good radiation enhancement effect during the physical and chemical lethal stage of radiotherapy. Furthermore, due to a long incubation time in biological stage, the 5-FU will give full play to its role in hindering the repair sublethal damage through its function of DNA synthesis interference [35, 67].”

2. Thank you for the suggestion, the loading capacity is very useful for us to know the loading amount of drugs in the NCs. Therefore, we have calculated the loading capacity of the two drugs in the NCs, which is 12.3% and 9.5% for MI and 5-FU respectively, and we have supplemented it in section 2.4 (page 4, section 2.4, line 10), and section 3.1 (page 7, last paragraph, last sentence) (LOA 3).

3. We have added the references of the cytotype dependence chemoresistance caused by hypoxia condition (page 11, last paragraph) (**LOA 6**).

Moreover, we have revised the discussion section as suggested (**LOA 5**). As summary, we have divided the discussion into seven parts, which are research background (page 10-11, section 4, first paragraph), antitumor effect of the NCs (page 11, paragraph 2), the additive effect between 5-FU and MI (page 11, paragraph 3), the cytotype dependence phenomenon (page 11-12, last paragraph), the discussion of hypoxia sensitivity results (page 12, paragraph 2), the radiotherapy improvement mechanism of the NCs (page 12-13, paragraph 3) and the limitations of the present study (page 13, paragraph 2) to make the discussion section more comprehensible to the readers.

Thank you for your comments which improve our revision a lot.

In addition, we are sorry for the typos in the rebuttal letter, and we have checked it carefully to minimize this error this time. Thank you for your careful review again.

4.5. Thank you for your work in the two review processes.

Lastly, we would like to thank you again for all your work and comments in these two reviews. Your suggestions are all valuable, constructive and instructive for us in the present and further studies. And we will be more careful and diligent in the future studies. Thank you very much.